# Cognitive–Affective Negotiation Process in Green Food Purchase Intention: A Qualitative Study Based on Grounded Theory

**DOI:** 10.3390/foods14162856

**Published:** 2025-08-18

**Authors:** Yingying Lian, Jirawan Deeprasert, Songyu Jiang

**Affiliations:** Rattanakosin International College of Creative Entrepreneurship, Rajamangala University of Technology Rattanakosin, Nakhon Pathom 73170, Thailand; lian.yingying@rmutr.ac.th (Y.L.); jirawan.dee@rmutr.ac.th (J.D.)

**Keywords:** green food, purchase intention, grounded theory, green trust, consumer behavior

## Abstract

Green food serves as a bridge connecting healthy lifestyles with environmental values, particularly in the context of sustainable consumption transitions. However, existing research lacks a systematic understanding of how consumers negotiate cognitive evaluations and emotional responses when forming green food purchase intentions. This study addresses that gap by exploring the cognitive–affective negotiation process underlying consumers’ green food choices. Based on 26 semi-structured interviews with Chinese consumers across diverse socio-economic backgrounds, the grounded theory methodology was employed to inductively construct a conceptual model. The coding process achieved theoretical saturation, while sentiment analysis was integrated to trace the emotional valence of key behavioral drivers. Findings reveal that external factors—including price sensitivity, label ambiguity, access limitations, social influence, and health beliefs—shape behavioral intentions indirectly through three core affective mediators: green trust, perceived value, and lifestyle congruence. These internal constructs translate contextual stimuli into evaluative and motivational responses, highlighting the dynamic interplay between rational judgments and symbolic–emotional interpretations. Sentiment analysis confirmed that emotional trust and psychological reassurance are pivotal in facilitating consumption intention, while price concerns and skepticism act as affective inhibitors. The proposed model extends the Theory of Planned Behavior by embedding affective mediation pathways and structural constraint dynamics, offering a more context-sensitive framework for understanding sustainable consumption behaviors. Given China’s certification-centered trust environment, these findings underscore the cultural specificity of institutional trust mechanisms, with implications for adapting the model in different market contexts. Practically, this study offers actionable insights for policymakers and marketers to enhance eco-label transparency, reduce structural barriers, and design emotionally resonant brand narratives that align with consumers’ identity aspirations.

## 1. Introduction

With the intensification of global ecological and environmental challenges, the concept of green development has become a core strategic approach to facilitating socio-economic transformation. Guided by national strategies such as the “Dual Carbon” goals, rural revitalization, and the Healthy China initiative, China is accelerating its transition toward a high-quality development model characterized by ecological prioritization and low-carbon growth [1,2]. As a vital pathway to achieving sustainable development, green consumption is increasingly integrated into national policies and governance frameworks. Among its components, green food plays a pivotal role, fulfilling multiple functions: on one hand, it represents the high-quality advancement of agricultural ecology, reflecting principles of clean production, resource conservation, and environmental protection; on the other, it aligns with evolving public health concerns, meeting consumers’ growing demands for nutritious, safe, and sustainable food products [3,4,5]. Green food is no longer merely an agricultural product, but increasingly serves as a symbolic driver of green lifestyles and a strategic element within China’s broader development agenda [6,7].

Green food refers specifically to food products that are certified by the China Green Food Development Center (CGFDC), following standards of limited chemical input, controlled environmental impact, and traceable ecological production systems. This definition aligns with the national standard NY/T 391–2021 and distinguishes green food from other labels such as organic or pollution-free food. Unlike organic food, which eliminates synthetic inputs entirely, or pollution-free food, which focuses solely on residue-free production, green food certification strikes a pragmatic balance by permitting controlled input usage under rigorous ecological management and process monitoring. The certification system includes routine audits, ecological index monitoring, and production process tracing to ensure credibility and uniformity across products.

Nevertheless, the market penetration and public adoption of green food remain relatively low. More than 55,000 products have been certified as green food nationwide [8]. However, their share in total food consumption has remained consistently below 5%, falling short of expectations for agricultural ecology. Consumer understanding of green food is often ambiguous, with many confusing it with similar concepts such as organic food or pollution-free food. Although over 60% of respondents express support for green consumption in principle, they frequently abandon green food purchases in practice due to concerns over higher prices, lack of trust, unclear labeling, and insufficient differentiation [9,10]. To a certain extent, this disconnect may be due to the lack of a clear motivational framework or a solid cognitive system to support the practice of sustainable consumption behavior. In this context, there is an urgent need for a more detailed insight into the psychological motivations and situational factors that affect the intention to purchase green food. Clarifying the internal motivations and external pressures that shape such intentions is of fundamental significance for promoting the further development of the green food market.

It is worth noting that in this field, consumers’ purchasing decisions are often not influenced by superficial factors such as price, labeling or supply alone. Instead, deeper driving factors—such as individual beliefs, consumption habits, self-concept and emotional investment—often play a key role. In addition, broader contextual forces such as government policies, family norms, and mass media discourse also have a significant impact on consumer preferences in a subtle way [11,12].

In recent years, a substantial body of research has been dedicated to examining the factors that influence green food consumption behavior, primarily through rationalist paradigms such as the Theory of Planned Behavior (TPB) and the Health Belief Model (HBM). The effect of individual characteristics and cognitive perceptions on green food purchasing behavior has been analyzed, with findings emphasizing the role of consumer awareness and personal values in intention formation [13]. Furthermore, the interaction between attitudes, habitual consumption practices, and lifestyle orientations has been explored to reveal key behavioral pathways using Structural Equation Modeling (SEM) and Necessary Condition Analysis (NCA) [14]. The influence of perceived usefulness, environmental problem salience, and pandemic-related fears on young consumers’ green purchase intentions has also been investigated to understand emerging motivational triggers [15].

While these studies offer valuable insights, they are largely rooted in variable-centric, quantitative methodologies that may not fully capture the emotional, symbolic, and situational complexities embedded in green consumption behaviors. Specifically, factors such as emotional trust in eco-labels, identity-driven motivations, and the dynamic negotiation of structural barriers (e.g., price sensitivity, access limitations) remain underexplored. Existing models tend to conceptualize intention formation as a linear outcome of cognitive evaluations, thereby overlooking the affective mediators and contextual resistance that often disrupt the translation of ecological attitudes into actual purchasing behavior. For instance, Mazhar and Zilahy [14] provided structural insights into habitual practices but lacked attention to emotional trust pathways; Zhang et al. [15] emphasized pandemic-triggered cognitive shifts without addressing how such fears are emotionally negotiated within peer-influenced identity constructs; and Deng et al. [13] acknowledged demographic heterogeneity but did not explore the symbolic meanings consumers assign to green food under institutional ambiguity.

Given these limitations, there is a pressing need for an inductive, emotion-aware research approach that can elucidate the micro-mechanisms through which consumers internalize external stimuli and negotiate behavioral intentions. Grounded theory methodology is particularly suited to this aim as it allows for the iterative development of theory grounded in rich empirical narratives, facilitating the discovery of emergent categories that extend beyond predefined variables. This study adopts a grounded theory methodology to construct a cognitive–affective negotiation model of green food purchase intention, thereby addressing the conceptual gaps identified in the recent literature and providing a more context-sensitive understanding of sustainable consumption behaviors.

## 2. Literature Review

### 2.1. Green Food Purchase Intention

Green food purchase intention refers to a consumer’s subjective willingness or likelihood of buying food products that are certified as environmentally friendly, safe, and sustainably produced [16], serves as a critical psychological precursor to actual green food consumption, and reflects an individual’s alignment with ecological values, health consciousness, and socially responsible behavior. Green food purchase intention is influenced by a complex set of factors, typically categorized as internal and external [17].

Table 1 outlines a set of twelve influencing factors, derived through a synthesis of foundational behavioral theories and recent empirical investigations. Notably, psychological variables on the individual level—such as one’s awareness of environmental issues, personal health orientation, subjective valuation of green food, and emotional connection—are repeatedly discussed within frameworks like the Theory of Planned Behavior (TPB) and the Health Belief Model (HBM). The TPB posits that behavioral intention is determined by attitudes toward the behavior, subjective norms, and perceived behavioral control. Attitudes reflect personal evaluations of green food consumption (e.g., its perceived benefits and ecological value), subjective norms capture the influence of social expectations and peer behavior, while perceived behavioral control pertains to individuals’ assessment of the feasibility of purchasing green food given external constraints. The HBM, in contrast, emphasizes health-related constructs such as perceived susceptibility (perceived risk of health problems), perceived severity (the seriousness of consequences), perceived benefits (advantages of green food consumption), and perceived barriers (obstacles like price and accessibility), providing a structured framework for analyzing health-driven motivations in green food purchasing.

However, while Table 1 offers a comprehensive taxonomy of influencing factors, existing studies predominantly adopt variable-driven quantitative designs, which limits the exploration of emergent, context-sensitive motivations. For instance, studies like Ganesh and Mohamed [22] validated cognitive variables such as consumer knowledge, but did not address how emotional attachment or identity-related motives influence consumption decisions. Similarly, Sheikh and Tourani [19] highlighted the importance of institutional trust without considering how such trust is emotionally constructed through personal experiences and social narratives. These methodological tendencies constrain the capacity to understand how internal dispositions and external pressures co-evolve in actual consumer behavior.

Recent research efforts have begun to address the evolving complexity of green food consumption behaviors by integrating broader behavioral constructs and contextual drivers. For instance, Mazhar and Zilahy (2025) explored how attitudes, habitual practices, and lifestyle alignment interact to shape green purchasing behaviors, revealing that such behaviors are not merely outcomes of rational deliberation but are deeply embedded within consumers’ routines and aspirational lifestyles [14]. Zhang et al. (2024) further advanced this perspective by highlighting the influence of pandemic-induced behavioral shifts, demonstrating that public health crises and environmental degradation serve as potent cognitive triggers for green consumption, particularly among young and environmentally conscious cohorts [15]. Additionally, Deng et al. (2024) emphasized the role of individual cognitive capacity and demographic characteristics in shaping green food consumption patterns, underscoring the heterogeneity of behavioral responses across different population segments [13].

Moreover, the interaction between internal and external factors is inherently complex and dynamic. For example, a consumer’s health motivation may enhance the perceived value of eco-labeled products, but simultaneously conflict with price sensitivity, resulting in a negotiation process between personal aspirations and economic feasibility. Similarly, institutional trust is not a static factor but is continually reinterpreted through consumers’ emotional trust frameworks, shaped by their past experiences and peer influences. Existing studies tend to examine these factors in isolation, thereby overlooking the nuanced negotiation process that consumers undertake when forming green food purchase intentions.

Despite these contributions, a significant gap remains regarding the emotional mechanisms through which consumers interpret, internalize, and negotiate the meanings associated with green food. Existing studies have predominantly focused on cognitive antecedents—such as environmental awareness, health consciousness, and perceived behavioral control—or on structural enablers like price and availability. However, the ways in which emotional trust, perceived value resonance, and symbolic lifestyle congruence function as affective mediators in the formation of purchase intention remain underexplored. This analytical omission limits the understanding of how consumers navigate the cognitive–affective negotiation process that underpins green consumption behaviors in real-world contexts.

In addition to theoretical limitations, practical barriers to green food adoption—such as low market penetration and widespread consumer confusion regarding eco-label distinctions—remain insufficiently explored. Although green food certification is well-defined within regulatory frameworks, many consumers conflate it with “organic” or “pollution-free” labels, leading to uncertainty about product authenticity and value. Studies have shown that this conceptual ambiguity, coupled with structural barriers like higher pricing and limited availability, significantly hinders the translation of pro-environmental attitudes into actual purchasing behaviors [9,10]. Addressing these real-world challenges requires analytical models that move beyond cognitive constructs to incorporate emotional assurance and contextual clarity.

Consequently, while the extant literature provides valuable insights into the antecedents of green food purchase intention, several critical limitations persist. First, the majority of current research adopts a quantitative, variable-centric approach grounded in established models such as the Theory of Planned Behavior (TPB) and the Health Belief Model (HBM). This design tends to pre-define influencing factors, thereby restricting the discovery of emergent, context-specific drivers that may arise from lived consumer experiences. Second, internal psychological constructs and external situational factors are frequently examined in isolation, without systematically exploring how they interact and co-evolve in dynamic consumption environments. Third, many studies concentrate on narrowly defined consumer segments—such as Gen-Z or urban middle-class consumers—thus failing to capture the diversity of the broader consumer base, especially in markets characterized by rapid cultural and economic transitions.

Moreover, the underlying mechanisms through which consumers interpret green food-related information, negotiate trust and authenticity, and reconcile external constraints with personal values remain insufficiently theorized. These analytical gaps are particularly pronounced in the Chinese context, where green food adoption is shaped by a complex interplay of social norms, emotional trust, institutional credibility, and market dynamics. Addressing these gaps requires a more integrative and context-sensitive analytical framework that can capture the nuanced interactions between cognitive stimuli and emotional pathways in green food consumption decision-making.

### 2.2. Theoretical Perspectives on Green Food Purchase Behavior

Understanding how consumers form their intention to purchase green food requires a theoretical framework that reflects a combination of rational judgment and normative considerations. The TPB is widely considered to be an important model for explaining green consumption behavior because it integrates key components such as attitudes, normative beliefs, and perceived behavioral control. Other influential theoretical models include the Technology Acceptance Model (TAM), Social Cognitive Theory (SCT), Health Belief Model (HBM), and Trust-Oriented Behavior Theory. These theories provide diverse conceptual perspectives for research in related fields and help identify a set of variables that have a significant impact on consumers’ green food purchasing behavior, such as individual attitudes, social norms, self-efficacy, perceived behavioral control, and interpersonal trust [33].

The TPB is also explanatory in the broader context of environmental decision-making. In high-risk scenarios such as disaster response, the model is often used in conjunction with the trust-oriented model to gain a deeper understanding of the behavioral response mechanism of individuals to group risks. It has been observed that both trust and perceived control significantly enhanced compliance behavior during flood evacuations [34]. Similar findings have been reported in prosocial domains, such as religious giving, where trust in institutional authority strengthened behavioral intention when TPB constructs were mediated by trust [35]. These studies underline that the integration of trust with the TPB provides a more nuanced understanding of intention formation, particularly where ethical or moral judgments are involved.

In addition to behavioral models, sociological perspectives have contributed to understanding how contextual and communicative factors interact with personal preferences. It has been noted that decision-making in areas such as transportation or food is not purely rational but is increasingly recognized as symbolically and emotionally mediated. In the domain of green consumption, emerging concepts such as lifestyle congruence—the perceived compatibility between one’s daily practices and green consumption demands—and affective trust, defined as emotionally grounded belief in authenticity and shared values, have gained growing theoretical attention [36,37]. These constructs extend the social–normative foundation of the TPB by emphasizing identity negotiation and emotional resonance as crucial pathways shaping intention. Cultural values, especially those that emphasize collective identity and social cohesion, may amplify these effects, reinforcing consumers’ alignment with symbolic norms and trust-based cues rather than formal institutional credibility alone [38].

However, existing research predominantly relies on variable-driven models such as the TPB and HBM, which predefine influencing factors and linear relationships, limiting the capacity to uncover emergent and context-specific mechanisms. In scenarios where consumers navigate complex situational constraints—such as price sensitivity, label ambiguity, and access limitations—the dynamic negotiation between emotional trust, identity-driven motivations, and structural barriers remains insufficiently explored. These intricacies are difficult to capture through pre-structured survey instruments or regression-based models. Therefore, this study adopts a grounded theory methodology, which allows for the inductive extraction of cognitive–emotional–contextual negotiation pathways from consumers’ micro-level narratives. This approach enables the identification of dynamic, adaptive decision-making mechanisms that conventional variable-based models are not equipped to reveal, thus addressing the theoretical and empirical gaps highlighted in the literature review.

Overall, the aforementioned theoretical frameworks collectively present the understanding that consumers’ purchase intentions are shaped by the continuous interaction between personal beliefs, perceived control, institutional trust, normative expectations, and social context. However, despite their conceptual significance, these theories still have certain limitations. A major criticism is that these models often rely on fixed variables and standardized scales, which may not fully capture behavioral drivers that are context-sensitive or dynamically generated. In addition, these theories often regard the relationship between individual tendencies and external influences as linear and static, ignoring the more complex, adaptive, or negotiated behavioral characteristics that people exhibit in real-life environments. By focusing on how consumers actively interpret and respond to green food choices in their real-life experiences, grounded theory can identify micro-motivational structures and behavior generation mechanisms that may be overlooked in traditional variable-driven analysis.

Despite the conceptual significance of existing theoretical frameworks, several critical gaps remain unaddressed. Firstly, prior studies have not systematically revealed the cognitive–affective–contextual negotiation pathways through which green food purchase intentions are formed in real-world decision-making environments. Secondly, the interactive mechanisms involving emotional trust, identity alignment, and structural constraints (such as price sensitivity and eco-label ambiguity) lack micro-level empirical explanation, resulting in a fragmented understanding of consumer decision logic. Thirdly, the behavioral dynamics of heterogeneous consumer groups—such as family decision-makers and rural–urban fringe consumers—have been underexplored, limiting the generalizability of existing models. In response to these gaps, this study adopts a grounded theory approach to construct a Cognitive–Affective–Contextual Negotiation Model of green food purchase intention. By integrating emergent themes derived from rich consumer narratives, this research aims to provide a process-oriented and context-sensitive analytical framework that bridges the current theoretical and practical voids.

## 3. Methods

This study employs a qualitative research design grounded in the principles of grounded theory, a methodology originally developed by Glaser and Strauss [39] that emphasizes theory generation from systematically collected and analyzed data. In the context of green food purchase intention, this methodological orientation is especially relevant. Grounded theory offers the flexibility to examine how consumers negotiate value conflicts, emotional ambivalence, institutional trust, and lifestyle alignment—all of which may influence their behavioral intentions in non-linear and socially embedded ways. Given the exploratory nature of this study and the objective of capturing both internal dispositions and external influences in a dynamic social context, grounded theory provides a suitable methodological foundation for developing an inductively constructed framework of green food decision-making.

Given the methodological evolution of grounded theory, this study deliberately adheres to Glaser’s classic discovery-oriented approach rather than the structured coding paradigm of Strauss and Corbin or the reflexive constructivist stance of Charmaz. That choice aligns with this study’s objective of inductively exploring the cognitive–affective negotiation mechanisms in green food consumption, as Glaser’s approach emphasizes theoretical emergence and analytical flexibility, which are essential for capturing context-specific and affect-driven behavioral dynamics without pre-imposed coding schemas. Unlike phenomenology, which seeks to describe lived experiences without necessarily building theory, or ethnography, which requires long-term cultural immersion, grounded theory is uniquely suited to uncovering process-oriented, emergent mechanisms from participants’ narratives. Given this study’s focus on how cognitive, emotional, and situational factors are dynamically negotiated in green food consumption—a phenomenon lacking predefined theoretical constructs—Glaser’s discovery-driven approach provides the necessary analytical flexibility.

To enhance transparency and traceability, the methodological logic of this study follows a phased structure aligned with the principles of grounded theory. The research process began with the identification of conceptual gaps through a literature review, followed by purposive and snowball sampling for semi-structured interviews. Data were analyzed using open, axial, and selective coding in NVivo 12. Theoretical saturation was continuously assessed, and coding reliability was ensured through double-coding and intercoder consensus. A visual summary of the research process is provided in Figure 1 to clarify the sequential steps from design to theory development.

### 3.1. Research Methods and Data Collection

This paper adopts a qualitative research method based on grounded theory, because this method is to support an open-ended research path to deeply understand how individuals form green food purchase intentions, especially focusing on multiple influencing factors from internal cognition and external environment.

Empirical data were collected through semi-structured interviews. Each interview focused on the attitudes, motivations, and situational influences on participants in the consumption decision-making process, to guide them to reflect deeply and express themselves in detail. In total, 26 interviews were conducted, and the interview format was flexibly selected according to the schedule and geographical distribution of the participants, including face-to-face communication and online video conferences. The duration of each interview ranged from 45 to 90 min. This time range ensured that the research subjects had sufficient time to express their diverse views and real experiences, which was in line with the interpretive research orientation of this study. With the informed consent of the participants, all interviews were recorded and transcribed verbatim to maximize the original appearance and richness of their narratives.

The interview outline (Appendix A) is designed to be highly flexible, aiming to ensure that core issues (such as perception of risks and benefits, price sensitivity, trust in the system, and mainstream social expectations) are covered while allowing researchers to guide in-depth discussions on emerging topics based on the actual situation at the interview site.

The sample recruitment combining purposive sampling and snowball recommendation. Initial participants were purposively selected based on their active involvement in household food purchasing and varying degrees of familiarity with green food concepts. Specific selection criteria included demographic diversity (age, occupation, location) and relevance to key research themes (e.g., health concerns, environmental awareness). Snowball sampling was guided by a sampling matrix that ensured referred participants expanded demographic and experiential diversity, thereby mitigating homogeneity bias. In principle, individuals who are closely related to green food consumption issues in terms of social roles, backgrounds, or life experiences are preferred. This study pays special attention to constructing heterogeneous samples to reflect the diverse psychological orientations and situational conditions in green consumption behaviors.

For example, the inclusion of parents and family caregivers in the study stems from their higher sensitivity to food safety and health issues, which helps to reveal the logic of consumption decisions based on health motivations [38]. Youth groups, especially college students and young professionals entering the workplace, are included because they are exposed to contemporary environmental issues, peer values, and market communication content and have research value for insight into identity-oriented behaviors [40]. In addition, participants from industries such as healthcare, education, and agriculture provided professional perspectives on institutional trust and professional responsibility [41], while respondents from rural and urban–rural areas reflected issues related to green food accessibility and skepticism about the environmental labeling system, which are often closely related to broader structural constraints. Table 2 summarizes the basic demographic characteristics and social context of the sample in this study.

The final sample size of 26 interviews was determined through the principle of theoretical saturation, consistent with the grounded theory methodology. Rather than predefining the number of participants, this research adopted an iterative approach, where data collection and coding were conducted simultaneously. As detailed in Section 3.3, theoretical saturation was observed after the 24th interview, when no new concepts or categories emerged from the data. The final two interviews confirmed this stability. This saturation-based sampling strategy ensures that the sample size is empirically grounded and conceptually sufficient to capture the complexity of green food purchase intentions.

The sampling framework of this study was designed not only to reflect the diversity of population characteristics but also more importantly to ensure the conceptual relevance of the sample, thereby facilitating the identification of psychological and social contextual mechanisms that influence green food purchasing behavior.

### 3.2. Interview Guide Design

In order to explore the underlying mechanisms that influence consumers’ intention to purchase green food, this study designed a semi-structured interview guide based on the grounded theory methodology. The content of the interview protocol was closely aligned with the research objective, focusing specifically on the internal psychological factors and external contextual conditions that affect purchase intention. The guide was intended to facilitate the systematic collection of rich qualitative data and to provide a structured basis for subsequent open, axial, and selective coding in theory development.

To ensure conceptual consistency, participants were briefly introduced to the nationally accepted definition of green food at the start of each interview. Prior to the formal interview, the researcher introduced the purpose of this study, clarified the ethical guidelines, and obtained written informed consent from all participants. Basic demographic and background information was collected at the beginning of each interview, including gender, age, occupation, educational background, household composition, and green food consumption experience. These data contributed to the analysis of sample diversity and theoretical saturation. The formal interview process was structured around four key dimensions: first, the participant’s perception and initial attitude toward green food; second, internal influencing factors, including health awareness, environmental values, emotional responses, and identity-related motivations; third, external influencing factors, such as price sensitivity, access to purchase channels, social influence, label trust, and institutional support; and fourth, the barriers and contradictions participants experienced in the transition from intention to action. At the end of each interview, open-ended questions invited participants to express any additional perspectives or experiences not yet covered.

The finalized interview guide used in this study is presented in Appendix A. To ensure both methodological rigor and theoretical sensitivity, the development of the semi-structured interview guide was informed by a combination of grounded theory principles and established behavioral models. Table 3 presents the theoretical rationale underlying each section of the interview framework. The development of the interview guide followed a two-stage iterative process. Initially, key constructs from the TPB, HBM, and trust theories were operationalized into open-ended questions addressing attitudes, normative pressures, and perceived barriers. A pilot round of three interviews was conducted to evaluate question clarity, narrative depth, and thematic coverage. Feedback from the pilot phase informed revisions that enhanced the guide’s flexibility, allowing respondents to elaborate on emotional drivers and contextual constraints. This iterative refinement ensured both theoretical alignment and empirical sensitivity in capturing emergent themes.

This research has been reviewed by the Research Ethics Committee of Maha chulalongkornrajavidyalaya University (Certificate No. R.160/2025). The Committee has affirmed that the research adheres to the International Code of Ethics, national laws, and regulatory requirements.

### 3.3. Validation Strategy

To ensure methodological rigor and analytical credibility, this study adopted Manuj and Pohlen’s [42] four trustworthiness criteria—credibility, transferability, dependability, and confirmability—and implemented them within the logic of the grounded theory methodology.

For dependability, a structured audit trail was maintained throughout this study. The use of NVivo 12.0 facilitated transparent and traceable data management, and the researcher engaged in memo writing to document evolving theoretical sensitivity, coding decisions, and interpretive reasoning, ensuring consistency across analytical stages. Transferability was enhanced by providing detailed contextual descriptions of participant demographics and socio-environmental settings (Table 2), enabling analytical generalization to similar consumption contexts. Confirmability was ensured through a transparent audit trail, including verbatim transcripts, coding logs, analytical memos, and decision audit forms, which were independently reviewed by a methodological auditor to minimize subjective bias.

To enhance coding reliability, the initial round of open coding was conducted independently by the lead researcher. A secondary coder—trained in grounded theory procedures—was later involved in reviewing a subset of the coded transcripts (n = 6) to ensure consistency in code application. Intercoder agreement was assessed through iterative discussion and consensus-building rather than statistical metrics, in line with qualitative standards. Discrepancies were resolved through dialogic reflexivity, and the codebook was refined accordingly. This dual-coding strategy improved analytical rigor while preserving theoretical sensitivity. To address the multi-layered complexity of consumer narratives, the coding process incorporated a dual-pathway strategy that systematically connected thematic categorization with emotional polarity mapping. This ensured that beyond the identification of cognitive categories, the emotional interpretations of key stimuli—such as trust, price, and institutional credibility—were empirically captured and traced through axial and selective coding stages. Such integration enhanced the depth and explanatory granularity of the emergent conceptual framework.

During open coding, line-by-line analysis yielded 213 initial codes such as “eco-label skepticism,” “price-induced hesitation,” and “peer-driven encouragement.” Axial coding aggregated these codes into higher-order categories by mapping causal conditions (e.g., “label confusion” linked to “institutional distrust”), contextual factors, and intervening variables. Selective coding identified “cognitive–affective negotiation” as the core category, integrating relational pathways between trust perceptions, value assessments, and situational constraints.

Theoretical saturation was assessed following the principles of grounded theory, specifically drawing on the constant comparative method [43]. Throughout data collection, interview transcripts were analyzed iteratively using open and axial coding. Saturation was operationalized as the point at which no new concepts, properties, or relationships emerged from additional data. By the 24th interview, coding redundancy began to appear across categories such as ‘green trust’, ‘price resistance’, and ‘label confusion’, indicating a stabilization of core themes. The final two interviews (25th and 26th) confirmed this stability, yielding no additional conceptual categories. A saturation grid was maintained, documenting new code emergence after each interview. Indicators for saturation included the absence of novel codes, repetition of existing conceptual linkages, and stabilization of core categories. By the 24th interview, no new properties emerged across primary themes, a finding validated through memo triangulation and intercoder consensus on coding redundancy.

Given this study’s primary objective of constructing a conceptual model to explain the cognitive–affective negotiation process of green food purchase intention, participant diversity was intentionally designed to enhance contextual coverage rather than to facilitate comparative subgroup analysis. Theoretical saturation was evaluated at the level of conceptual category stability across diverse situational narratives, not by numeric balance within demographic strata. Core categories such as ‘green trust’, ‘perceived value’, and ‘behavioral resistance’ consistently re-emerged across participants regardless of subgroup affiliation, indicating that the cognitive mechanisms underpinning purchase intention were sufficiently captured without necessitating proportional subgroup expansion. To enhance credibility, saturation assessment was corroborated through memo writing, codebook stabilization, and intercoder review of six randomly selected transcripts. This inductive stopping rule ensured that data collection was concluded not arbitrarily but based on empirical saturation, consistent with the grounded theory methodology.

### 3.4. Sentiment Analysis Procedures

To complement the open coding process and enhance the understanding of respondents’ affective tendencies, a sentiment analysis was conducted using NVivo 12’s auto-coding feature in conjunction with a curated Chinese sentiment dictionary derived from the Dalian University of Technology’s sentiment lexicon (DLUT-Emotion). The dictionary includes more than 27,000 Chinese emotional words categorized into positive and negative polarities, with sub-categories such as trust, fear, joy, and disgust. We manually verified and adjusted the polarity of ambiguous terms to reflect context-specific usage. Respondent statements extracted from the interviews were analyzed to calculate frequency-weighted emotional valence. The results were further triangulated with open codes such as “trust in certification,” “label confusion,” and “identity alignment,” allowing us to systematically map affective tendencies onto emerging theoretical constructs. This process helped validate the cognitive–affective dimension of the proposed model.

The sentiment analysis was conducted as an integral part of the grounded theory process, applied to the conceptual categories identified during open coding. Each extracted keyword represents a distilled expression of participant narratives, which were systematically coded into categories such as ‘health motivation,’ ‘emotional trust,’ ‘label ambiguity,’ and ‘price concern.’ By applying sentiment polarity mapping to these categories, we aimed to validate how cognitive cues are emotionally interpreted by consumers, thereby elucidating the affective mediating pathways (e.g., green trust, perceived value) that emerged in the axial and selective coding stages.

## 4. Results

### 4.1. Open Coding

Open coding is the basic stage in grounded theory analysis. Its core task is to systematically disassemble, carefully review, and label qualitative data. The goal of this process is to directly construct a preliminary category system based on the narrative content of the interviewees [44]. A total of 213 initial codes were generated in this stage. These codes were continuously refined and aggregated through the continuous comparison method in the subsequent process, and finally formed higher-level conceptual categories. These derived categories cover the internal psychological mechanisms and external situational factors that affect consumer decision-making. Following the principle of “theoretical sensitivity” in grounded theory, all categories are generated inductively rather than pre-determined.

The coding process ultimately identified ten core categories: (1) health motivation, (2) emotional trust, (3) environmental attitude, (4) price concern, (5) accessibility and availability, (6) label trust and confusion, (7) social influence, (8) policy and media messaging, (9) behavioral barriers, and (10) identity and value alignment. These categories cover key themes that recur in the respondents’ narratives and reflect the multidimensional characteristics of green food consumption behavior.

Health motivation was one of the most prominent themes, especially among participants with caregiving roles. Expressions such as “children’s health”, “food safety”, and “disease prevention” frequently appeared, indicating the importance of family health needs in green food selection.

These patterns are illustrated with selected participant quotes below, along with their corresponding initial codes: One mother shared, “My son has a sensitive stomach. I only trust food with the green label—it gives me peace of mind.” → This was abstracted into the initial code “children’s dietary safety”, representing health-driven consumption.

Another participant stated, “Even though it’s more expensive, I always choose green food for my family’s well-being.” → This corresponds to “peace of mind from green labels”, reflecting emotional reassurance in health-related decision-making.

To maintain structural clarity and avoid redundancy, this section selectively presents illustrative quotes from the ‘Health Motivation’ category as a representative example of how initial codes were derived from participant narratives. While direct quotes for the remaining nine categories are not exhaustively enumerated, they follow the same inductive coding logic, ensuring analytic consistency across all thematic constructs.

In terms of internal values, environmental attitudes were particularly prominent in the responses of young groups and urban residents. Expressions such as “supporting sustainability” and “low-carbon living” reflect personal recognition of environmental responsibility. Similarly, identity and value recognition were also presented as symbolic dimensions of green consumption behavior, with some respondents saying that such consumption “fits my lifestyle” or “reflects my emphasis on health.”

At the same time, this study also identified a series of behavioral barriers. Price concerns were the most frequently mentioned negative factor, with many participants expressing the view that “budgets are limited” or that green food is “not cost-effective.” In rural and urban–rural areas, accessibility issues were particularly prominent, with respondents saying that green food was “difficult to buy locally” or “can only be found in large supermarkets.” In terms of labeling, confusion between concepts such as “green,” “organic,” and “harmless” also led to a certain degree of hesitation and distrust, which was often summarized as “uncertainty of authenticity.”

In addition, social networks and institutional communication also play a role. Keywords such as “family recommendation” and “friends’ advice” show the influence of interpersonal relationships on consumer decisions, while government advocacy and media publicity are also seen as important channels for influencing cognition, although their credibility varies from person to person.

Finally, the category of “behavioral barriers” reveals the disconnect between consumers’ expressed purchase intentions and actual actions. Statements such as “I wanted to buy but didn’t” and “I gave up because it was inconvenient” show that even if the attitude is positive, practical limitations and psychological concerns in reality still hinder the adoption of green food.

The aggregation process from initial codes to core categories involved iterative comparison and thematic clustering. For instance, participant expressions such as “children’s dietary safety”, “elderly health concerns”, and “family well-being” were initially coded as distinct nodes. Through constant comparison, these were subsumed under the higher-order category of ‘Health Motivation’. Similarly, codes like “unreliable labeling” and “confusion with organic standards” were grouped to form the ‘Label Trust and Confusion’ category. This systematic abstraction process ensured that conceptual categories accurately encapsulated the shared patterns emerging from diverse participant narratives.

Overall, the open coding stage (Table 4) laid a solid empirical foundation for subsequent analysis and systematically distinguished the psychological, social and structural factors involved in the purchase intention of green food. These initially constructed core categories will serve as the logical starting point for analyzing the relationship between variables in the axial coding stage.

**Table 4 foods-14-02856-t004:** Open coding results: Conceptual categories and extracted keywords.

Preliminary Category	Representative Expressions/Extracted Keywords from Participant Narratives
Health Motivation	Child’s health
	Elderly dietary safety
	Peace of mind
	Health responsibility
	Disease prevention
Emotional Trust	Trustworthy label
	Packaging gives assurance
	Intuitive choice
	Green logo means safety
	Psychological comfort
Environmental Attitude	Environmental awareness
	Low-carbon lifestyle
	Support for sustainability
	Avoid over-packaging
	Doing something for the environment
Price Concern	Too expensive
	Cannot afford
	Much pricier than regular food
	Limited budget
	Not worth the price
Access and Availability	Hard to find
	Not available in small cities
	Only in major supermarkets
	Unreliable online sources
	Inconvenient to purchase
Label Trust and Confusion	Too many labels to understand
	Hard to tell authenticity
	Unclear what green food really is
	Confused with organic food
Social Influence	Recommended by family
	Tried it because friends did
	Curious after colleagues started buying
	Children requested it
Policy and Media Messaging	Saw it in public campaigns
	Media says it is healthier
	Government-endorsed brands are more trustworthy
	Influenced by promotional messages
Purchase Barriers	Intended to buy but could not find it
	Gave up because of price
	Uncertain about authenticity
	Did not buy due to skepticism
Identity and Value Alignment	Green consumption as responsibility
	Matches my lifestyle
	Shows I value health
	Reflects my environmental beliefs

Figure 2 aggregates high-frequency keywords and conceptually significant phrases identified during the open coding phase, such as “health,” “trust,” “price,” “environmental awareness,” and “value alignment.” These terms reflect the recurring themes extracted from 213 initial codes and grouped into ten preliminary categories, including health motivation, emotional trust, environmental attitude, price concern, and social influence. The visual emphasis in the word cloud not only reinforces the prominence of certain factors—such as perceived health benefits and price-related constraints—but also complements the conceptual framework developed through systematic coding. The relative size and visual significance of keywords correspond to their frequency of occurrence and conceptual weight in the overall data, thus intuitively showing the cognitive and situational dimensions that influence green food purchase intention.

In addition to thematic categorization, sentiment polarity mapping was applied to these categories to capture the emotional undercurrents embedded in participant narratives. For instance, categories such as ‘health motivation’ and ‘identity alignment’ predominantly reflected positive emotional expressions, reinforcing their roles as affective drivers in the formation of green purchase intention. Conversely, categories like ‘price concern’ and ‘label confusion’ exhibited strong negative sentiment, illustrating their function as emotional inhibitors within the cognitive–affective negotiation process.

In addition to thematic categorization, sentiment polarity mapping was applied to the open coding categories to capture the emotional undercurrents embedded in participant narratives. As shown in Figure 3, the sentiment polarity heatmap visualizes the emotional valence of key keywords, illustrating how specific cognitive cues are emotionally interpreted by consumers. Categories such as ‘health motivation’, ‘identity alignment’, and ‘environmental values’ predominantly exhibit positive emotional expressions, reinforcing their function as affective enablers that strengthen green purchase intention. Conversely, categories like ‘price concern’, ‘label confusion’, and ‘behavioral resistance’ display strong negative sentiment polarities, identifying them as emotional inhibitors that constrain the translation of ecological attitudes into actual purchasing behavior.

This polarity mapping plays a crucial analytical role by empirically validating the cognitive–affective negotiation mechanism underpinning green food purchase intention. It demonstrates how emotional responses—such as trust, reassurance, frustration, or skepticism—mediate the evaluative and motivational processes that shape consumers’ behavioral decisions. These sentiment patterns provided foundational evidence for constructing key emotional mediators (i.e., green trust, perceived value, and lifestyle congruence) in the axial and selective coding stages, enabling a deeper understanding of how external stimuli are internalized into behavioral intention pathways.

Beyond reinforcing positive affective drivers, the sentiment analysis also illuminated key emotional inhibitors that disrupt intention formation. Categories such as ‘Price Concern’, ‘Label Confusion’, and ‘Behavioral Resistance’ exhibited high-intensity negative sentiment polarities. Expressions like “too expensive to sustain” and “uncertain about label authenticity” reflect frustration and skepticism, which act as emotional barriers impeding the conversion of positive ecological attitudes into actual purchasing behavior. These negative emotional responses underscore the dual role of cognitive stimuli, wherein external constraints evoke affective dissonance that must be resolved through trust restoration or contextual adaptation.

An overall sentiment distribution analysis revealed that 62.2% of coded keywords reflected positive emotional tendencies, while 37.8% exhibited negative emotional responses. This proportion underscores that despite encountering situational barriers, consumers’ narratives are generally guided by health-driven motivations and value-based alignment, which serve as primary affective drivers of sustainable consumption behavior. Therefore, the sentiment analysis not only complements the grounded theory coding but also enhances the analytical granularity by delineating how emotional interpretations intersect with the cognitive pathways of green food purchase intention.

### 4.2. Axial Coding

Axial coding represents a critical analytical phase in grounded theory, wherein the open coding categories are reorganized into higher-order conceptual themes that reveal the structural, cognitive, and affective dynamics of the studied phenomenon [43]. Based on constant comparison and theoretical clustering, this study identified ten axial themes that capture the internal and external mechanisms underlying green food purchase intention. These themes include health motivation, environmental values, institutional trust, label uncertainty, perceived value judgment, experiential immersion, social referencing, policy-induced cognition, availability constraints, and behavioral resistance. Each theme reflects the key path that consumers go through in constructing, evaluating, and weighing green food purchase intentions.

At the level of internal motivation, two interrelated but distinctive themes emerged. Health motivation reflects the individual’s high attention to physical health, especially in the context of taking care responsibilities. Keywords such as “children’s health” and “elderly food safety” are frequently mentioned. At the same time, environmental values also occupy an important position in the interviewees’ narratives. Such values are usually manifested as personal commitment and normative responsibility for ecological protection, and they are often expressed through behavioral performance such as “low-carbon lifestyle” and “personal protection of the environment”. These two types of value orientations together constitute the intrinsic motivational basis for consumers to participate in green food consumption behavior.

Another prominent analytical dimension is consumers’ evaluation of institutional trust, especially the trust related to product labels. Some respondents expressed “intuitive trust” in green labels and believed that familiar logos or packaging could provide some psychological reassurance, but this trust was not consistent. In many cases, the trust evoked by labels was weakened by the vague boundaries between terms such as “green”, “organic”, and “harmless”, which in turn led to label uncertainty. This dual perception shows that the effect of institutional communication has dual characteristics: on the one hand, credibility can promote consumer confidence, and on the other hand, ambiguity may constitute a psychological barrier.

The third theme involves consumers’ perceived value judgment of green food. At this level, respondents tend to compare the positive meaning of green food in terms of morality or health with its economic cost. Although most people recognize its positive attributes, expressions such as “too expensive” and “not worth the price” also appear frequently, reflecting that price is still an important factor restricting consumer behavior.

In addition to rational cost–benefit analysis, some respondents also mentioned the experiential immersion in the process of green food consumption. They pointed out that when the purchase activity takes place in farmers’ markets or environmental theme stores, it is easier to bring a pleasant and immersive experience. This interaction of senses and emotions seems to enhance consumers’ psychological satisfaction and deepen their behavioral expression of values and identity.

At the social context level, there are two main axis codes worth emphasizing. The first is social referencing, that is, the influence of family members, friends, or other interpersonal networks on consumption decisions, which is often expressed as “my family recommends” and “my colleagues are buying”, reflecting normative encouragement or habitual following. The second is policy-induced cognition, which refers to the shaping effect of government advocacy or expert authoritative discourse on consumer cognition. Some respondents clearly pointed out that national-level propaganda or endorsement by authoritative institutions has a key impact on their perception of the authenticity and desirability of green food.

Finally, this study also revealed multiple structural and psychological barriers that restrict consumer behavior transformation. Structural problems (availability constraints) include geographical restrictions, insufficient supply in small markets, and poor distribution channels for green food. Psychological barriers (behavioral resistance) are reflected in the psychological distance between intention and behavior, and their roots include distrust, price sensitivity, or the perception of the complexity of the purchase process. These resistances are often expressed as statements such as “I wanted to buy it but gave up later” or “I’m not sure if this is formal”, reflecting the intersection of external and internal obstacles in the path of consumer behavior conversion.

Taken together, these themes reveal the complex generation mechanism of green food purchase intention, which not only involves the mobilization of individual values and emotional responses but is also subject to multi-dimensional constraints of social relations and the structural environment.

The interrelations among axial themes further illuminate the dynamic negotiation process of green food purchase intention. For example, ‘Health Motivation’ often amplifies ‘Green Trust’ by reinforcing consumers’ emotional reassurance in product safety, whereas ‘Label Uncertainty’ undermines this trust, generating behavioral hesitation. Similarly, ‘Environmental Values’ interact with ‘Perceived Value Judgment’, where moral commitments to sustainability enhance the perceived worth of green food, but are frequently counterbalanced by price sensitivity. These interactive pathways elucidate how cognitive evaluations, emotional investments, and situational realities coalesce into complex behavioral decisions.

Together, these ten axial themes provide a comprehensive and theoretically saturated framework that connects motivational values, emotional and cognitive trust structures, evaluative perceptions, experiential engagement, social framing, and structural limitations. As outlined in Table 5, each axial theme synthesizes specific open coding categories and offers a conceptual foundation for the subsequent selective coding phase, where the integrative theoretical model of green food purchase intention is constructed.

### 4.3. Selective Coding

Selective coding is the final stage of grounded theory analysis, which aims to integrate the conceptual categories identified by open coding and axial coding and construct a unified explanatory theoretical framework. The core goal of this stage is to identify a core category that can penetrate the internal and external dimensions of the research phenomenon, so as to achieve the clarification of theoretical logic and the saturation of empirical materials [43].

The core concept finally identified in this study is the cognitive–affective negotiation process of green food consumption intention. This study found that consumers’ behavioral intention is not a linear response to a single stimulus, but is based on the continuous interweaving of rational evaluation, emotional resonance, and situational constraints. This negotiation process is jointly influenced by multiple factors, including health concerns, lifestyle adaptation, information framework setting, and broader structural constraints, which jointly constitute the cognitive basis of sustainable consumption decisions. For instance, a young professional participant described her decision-making as follows: “I always feel green food is the right choice, but when the price is double, I hesitate. If it’s recommended by my close friends or I see it in eco-friendly stores, I’m more likely to buy despite the price.” This narrative exemplifies the cognitive–affective negotiation process, where initial positive attitudes are mediated by emotional trust and peer influence, yet challenged by price-related constraints. Such micro-level negotiations were recurrent across diverse respondent profiles, validating the centrality of this core category.

Based on selective coding, this study further summarized the ten key variables into two interrelated dimensions: external situational stimuli and internal interpretive mechanisms. The former includes flow-based engagement, health cognition, social referencing, pricing sensitivity, ambiguity in labeling, restricted availability, and attitudinal resistance. These variables are situational inputs, and their specific behavioral effects depend on their interactive relationship with individual cognitive and motivational systems.

For example, the immersive participation experience (flow) and health-oriented beliefs are often regarded as key factors in enhancing emotional significance and behavioral investment, which can strengthen consumers’ positive behavioral commitment in green consumption situations. In contrast, factors such as economic pressure, label confusion, and channel restrictions constitute realistic obstacles to behavioral realization and reduce the perceived feasibility of behavior. In addition, resistance behaviors such as distrust or hesitation caused by transaction complexity further exacerbate the gap between intention and behavior. As a situational variable, the effect of social influence depends on whether it is consistent with consumers’ normative expectations and relationship trust. It may produce positive reinforcement or only have a neutral effect.

The above external variables are transformed into behavioral motivations with psychological driving force through a set of internal regulatory mechanisms. Specifically, this study identified three key internal structures: green trust, perceived value, and lifestyle congruence. Green trust in this study reflects a dual-layered construct encompassing both institutional confidence (e.g., belief in certification, labeling systems, and official endorsements) and affective assurance (e.g., psychological comfort and intuitive sense of safety). It captures how consumers internalize the credibility of green food through both rational judgment and emotional association, especially in contexts where label ambiguity and institutional inconsistency coexist; perceived value combines functional evaluation with symbolic or ethical judgment; and lifestyle congruence measures the extent to which green food consumption resonates with consumers’ identity expression, values, and moral commitments.

In summary, the selective coding stage constructed a behavioral explanation model with high theoretical integration and a clear structure, which provides a solid theoretical basis for understanding the cognitive–emotional synergy mechanism behind green food purchase intention. Green trust encompasses institutional credibility and emotional reassurance, indicating consumers’ belief in the authenticity and safety of green food. Perceived value integrates functional assessment with symbolic and ethical evaluations, while lifestyle congruence in this research refers to the perceived alignment between green food consumption and an individual’s everyday identity, moral orientation, and aspirational lifestyle patterns. Rather than being purely rational, this construct embodies a symbolic dimension of consumption, whereby individuals use green food to express environmental commitment, health consciousness, and a sense of personal responsibility in their social environment.

In this study, green trust is defined as consumers’ emotional confidence in the authenticity, safety, and ecological integrity of green-labeled products, established through direct interactions with product-related cues such as eco-labels and packaging symbols. In contrast, institutional trust refers to consumers’ cognitive trust in the credibility of external authorities—including government agencies, certification bodies, and media—that govern and endorse green certification systems. While institutional trust shapes the perceived credibility of green certifications at a systemic level, green trust operates at the product level as an affective mediator that translates this perceived credibility into emotional reassurance and purchasing motivation. Thus, institutional trust functions as a contextual antecedent that influences the formation strength of green trust, but does not independently mediate purchase intention within the model’s framework.

Together, these ten variables explain how green food purchase intention emerges from the integration of contextual stimuli and internal valuation. The selective coding framework thus provides a theoretically grounded and empirically validated structure for understanding sustainable consumption behavior. The definitions of each variable are presented in Table 6.

Notably, this study also encountered deviant cases that diverged from the dominant cognitive–affective negotiation patterns. For example, a rural participant expressed complete disregard for green labels, stating, “I don’t trust any label; I only believe what I grow myself.” Such cases highlight boundary conditions where extreme skepticism towards institutional systems nullifies the mediating role of green trust, indicating that in contexts of pervasive institutional distrust, external stimuli fail to trigger the intended affective or evaluative pathways. These outliers reinforce the model’s explanatory scope while delineating its contextual applicability limits.

### 4.4. Model Construction and Theoretical Validation

Drawing upon the results of selective coding, this study constructed a theoretical model that explains the formation of green food purchase intention through the interplay between cognitive stimuli and affective mechanisms. The model integrates twelve variables, systematically categorized into cognitive antecedents and affective mediators.

The model adopts a two-layer structure to represent the cognitive–affective negotiation process underlying green food consumption decisions. The first layer consists of cognitive variables divided into three categories—positive, neutral, and negative—based on their directional influence on consumer behavior tendencies [45,46,47,48,49]. The second layer contains affective mediators that channel these cognitive stimuli into evaluative, symbolic, and emotional responses. These mediators—green trust, perceived value, and lifestyle congruence—function as internal mechanisms that translate external situational cues into purchase motivation [50,51].

The visual structure of Figure 4 further illustrates how cognitive antecedents, categorized by sentiment polarity (positive, neutral, negative), interact with affective mediators to shape green food purchase intention. Positive cognitive cues, such as flow experience and health beliefs, stimulate emotional pathways by enhancing green trust and reinforcing perceived value, thereby facilitating a favorable alignment with consumers’ sustainable lifestyle aspirations. Conversely, negative cognitive stimuli, including label ambiguity, price sensitivity, and access limitations, diminish the perceived feasibility and emotional coherence of green food consumption, acting as inhibitors that disrupt lifestyle congruence and dampen behavioral intention.

Social influence, positioned as a neutral cognitive factor, exerts a contingent effect that varies depending on normative alignment and interpersonal credibility. It can either amplify trust-based pathways when congruent with consumers’ value systems or attenuate motivational responses when perceived as externally imposed or inconsistent. By explicitly mapping these cognitive–emotional interactions, the model delineates a dynamic negotiation process wherein contextual stimuli are affectively interpreted and internalized, forming the evaluative and motivational basis for sustainable consumption behavior. This framework extends beyond linear rationalist paradigms by embedding sentiment-driven pathways that capture the nuanced interplay between cognitive stimuli, emotional resonance, and behavioral intention formation.

Rather than viewing intention formation as a linear and rational outcome, this model conceptualizes it as a dynamic negotiation process. Consumers interpret and internalize contextual information not only through cognitive assessment but also through emotional resonance, symbolic identity, and affective compatibility. As such, the model offers a more nuanced account of intention formation, particularly in settings where structural constraints and institutional ambiguities coexist with rising ecological awareness.

To ensure its theoretical robustness, the model underwent iterative validation across three dimensions: (1) saturation testing to ensure conceptual completeness; (2) consistency verification through constant comparison logic; and (3) triangulation with established theoretical frameworks including the Theory of Planned Behavior [52], Value–Belief–Norm Theory [53], and behavioral models centered on trust and value [54,55]. While these classical models emphasize rational decision constructs such as attitudes, norms, and perceived control, the present model extends them by embedding affective mediation and lifestyle alignment as core pathways of intention generation.

Importantly, this model responds to persistent limitations in prior research that often marginalize emotional and structural factors. By explicitly incorporating constraint-related variables such as label ambiguity, price sensitivity, and access limitation, alongside emotional constructs such as trust and value alignment, the framework provides a context-sensitive explanation of why ecological attitudes do not consistently translate into action. It also highlights the contingent role of social influence, which may either reinforce or weaken behavioral motivation depending on normative congruence and interpersonal credibility.

In sum, the model proposed in this study captures the multi-level dynamics of green food purchase intention. By bridging cognitive evaluation with emotional trust and symbolic alignment, it contributes a theoretically grounded, empirically saturated, and contextually adaptive framework for understanding sustainable consumption behavior.

## 5. Discussion

### 5.1. Theoretical Implications

This study adopts a grounded theory approach to uncover how green food purchase intention is shaped through a dynamic cognitive–affective negotiation process, rather than as a linear outcome of predefined variables. By embedding emotional trust, perceived value resonance, and contextual constraints into the analytical framework, this model transcends the limitations of traditional rationalist paradigms. It reconceptualizes intention formation as an adaptive negotiation between internal motivations and external stimuli, highlighting the situational, iterative, and emotionally situated nature of sustainable consumption decisions.

Traditional models such as TPB and HBM [45,55] primarily conceptualize green food consumption intention through stable attitudinal pathways or health-risk assessments. However, these frameworks often fail to capture the nuanced interplay between emotional trust, identity signaling, and situational resistance that shapes real-world decision-making. The present model extends these paradigms by embedding emotional mediators as central conduits through which external conditions—such as institutional credibility, social norms, and structural constraints—are cognitively interpreted and affectively internalized into behavioral intentions.

Rather than treating intention formation as a linear function of discrete variables, this model emphasizes the mediating role of cognitive–affective negotiation mechanisms. Emotional trust, perceived value assessments, and lifestyle congruence operate as interpretive filters through which consumers negotiate between aspirational motives and contextual constraints, thus shaping their final purchase intentions in a dynamic, context-sensitive manner.

Among these mediators, green trust functions as a dual-dimensional construct, encompassing both institutional trust (such as confidence in eco-certification systems) and emotionally grounded psychological reassurance (such as feelings of safety and identity alignment in food choices) [56]. Perceived value reflects a moral–economic trade-off, where ethical utility is weighed against financial cost [45], while lifestyle congruence relates to symbolic consumption and the alignment between green food and consumers’ aspirational self-identities [57]. These distinctions reveal that emotional mediators are not homogeneous but instead operate through differentiated symbolic and affective routes, each embedded in context-specific meaning systems. While prior studies have acknowledged the influence of emotional resonance on pro-environmental behavior [56,58], they often marginalize these factors as peripheral enhancers rather than core explanatory constructs. The present study challenges this view by positioning emotional trust and symbolic alignment as pivotal mediators, which actively transform external stimuli into sustainable consumption intentions. In doing so, it advances the theoretical discourse beyond the TPB’s rationalist assumptions, aligning with emerging behavioral models that emphasize the fluid, affect-laden nature of consumption decisions in complex socio-cultural contexts.

The theoretical model proposed in this study is further supported by recent empirical findings from sustainable consumption research in Western contexts. Clear ecological information, such as production transparency or eco-labeling systems, enhances consumers’ perceived trust and facilitates the activation of affective–cognitive pathways toward behavioral intention [59]. These effects are particularly salient in contexts where symbolic cues—such as label reliability or ethical production indicators—serve as signals of institutional credibility. Furthermore, the relevance of trust mechanisms is reinforced by studies showing that Eco-score systems significantly reduce consumer skepticism and enhance intention through improved informational clarity and label credibility [60]. These findings validate our model’s emphasis on green trust and institutional confidence as pivotal emotional mediators. In addition, research on pre-consumption psychological states such as expected satiety has demonstrated that perceived behavioral feasibility—especially in terms of anticipated satisfaction and functional adequacy—also contributes to the intention formation process, thereby confirming the significance of structural and affective anticipations within sustainable food decisions.

It is worth emphasizing that emotional dimensions are often marginalized in previous studies, while this study systematically embeds them into the explanatory framework. Although some studies have mentioned the role of emotional resonance in behavioral motivation [56,58], few models formally incorporate such variables into structural analysis. The results of sentiment analysis in this study further validate the importance of emotional factors—more than 60% of respondents expressed emotional fit through constructs such as trust, identity consistency, and sustainable commitment. By incorporating emotional trust and lifestyle consistency into the model, this study reveals their synergistic effects in strengthening psychological value and behavioral intention.

Moreover, this framework demonstrates how different consumer groups activate these mechanisms in divergent ways. Parents and caregivers emphasize health-related trust and label reliability [38]; young adults tend to construct purchase intention through identity expression and peer validation [39]; while rural consumers often encounter structural barriers such as access limitations and price sensitivity, which reduce their behavioral feasibility despite positive attitudes [40]. These heterogeneous patterns illustrate that the intention formation process is deeply embedded in role-specific and context-sensitive environments. The primary theoretical contribution of this study lies in its articulation of a cognitive–affective negotiation framework, which captures how consumers reconcile aspirational motives with situational barriers through iterative emotional–cognitive filtering. By foregrounding affective mediators as structural mechanisms—rather than post hoc modifiers—the model offers a more holistic explanation of the intention–behavior discrepancy in green food consumption. This process-oriented, sentiment-anchored perspective provides a critical extension to existing frameworks by addressing how trust erosion, label ambiguity, and identity alignment coalesce within real-life decision environments.

In addition, this study highlights the impact of structural constraint-based variables, which have been often overlooked in the past. Price concerns, unclear labels, limited availability, and resistance based on suspicion—although mostly treated as control variables or peripheral constraints in existing models—are shown here to significantly inhibit purchase intentions even among individuals with high ecological awareness and positive attitudes. This finding helps explain the long-standing “intention–behavior discrepancy” in sustainable consumption and highlights the moderating power of the external context on behavioral transformation.

Although this study is based on the social and cultural context of contemporary China, its theoretical concerns respond to a broader academic gap. Although Balakrishnan and Dwivedi [57] pointed out the importance of consumer cognitive awareness, they failed to systematically explore the interactive mechanisms of trust, emotion, and identity in the process of intention construction. In contrast, this paper introduces a variety of social psychological variables and constructs a more integrated and context-sensitive explanatory model, which better reflects the complexity of real consumer behavior.

Additionally, by integrating grounded theory with sentiment analysis, this study offers a novel methodological approach that bridges qualitative theorization with data-driven emotional mapping [56,58]. This dual-layered method not only enriches the affective dimension of behavioral modeling but also facilitates a more empirically grounded, narrative-sensitive understanding of how consumers construct purchase intentions under complex socio-structural influences.

### 5.2. Practical Implications

The empirical findings of this study suggest that green food manufacturers should shift their strategic focus from narrow product functional appeals to the in-depth construction of emotional meaning and symbolic resonance. Rather than relying solely on technical indicators (such as safety certifications and eco-labels) for promotion, companies should focus on building relational trust and narrative-based brand identity. However, shifting from functional appeals to emotional resonance is not without challenges. One key difficulty lies in the cognitive ambiguity that still persists among consumers regarding eco-label definitions, which weakens the effectiveness of emotional narratives if not aligned with informational clarity. Additionally, emotional branding strategies require long-term engagement and resource investment, which may be difficult for small and medium-sized enterprises (SMEs) to sustain. To overcome these challenges, firms should adopt a phased communication approach that initially anchors emotional narratives in verified product transparency (e.g., behind-the-scenes production stories) and progressively builds symbolic resonance through community-based brand advocacy and peer-endorsed testimonials.

Consumers’ purchase intention is closely related to their emotional trust in products, their perceived congruence with personal values and lifestyle norms, and the emotional consistency conveyed in brand communication. Therefore, green food brand strategies should go beyond the traditional rational appeal model and pay more attention to the expression paths of authenticity, narrative richness, and identity alignment. In practice, improving label transparency, incorporating emotional brand narratives, and conveying sustainability concepts through sensory packaging design can all help build lasting consumer–brand relationships. Focusing on strategic communication of symbolic framing and lifestyle positioning can also help enhance brand attachment and promote long-term behavioral continuity. Moreover, consumer heterogeneity necessitates differentiated branding strategies. For family caregivers, brand messaging should prioritize health-related assurances and emotional trust in product safety, while for young, urban consumers, identity-driven narratives and peer-influenced engagement are more effective. In rural markets, affordability framing and access-driven incentives should be combined with localized storytelling that resonates with community values and practical concerns. Recognizing these divergent motivational pathways is critical to ensuring that branding efforts achieve resonance across diverse consumer segments.

From an institutional perspective, this study underscores the critical importance of systemic trust and infrastructural support in advancing green food adoption. While most consumers demonstrate favorable attitudes toward green consumption, their behavioral conversion remains significantly hindered by persistent structural barriers—such as pricing constraints, label ambiguity, and limited access to certified products. This discrepancy reflects a pronounced gap between pro-environmental intentions and actual behavioral feasibility. However, policy interventions face several practical challenges. For example, green food subsidies may face administrative bottlenecks and budget constraints, while regional distribution platforms may struggle with logistical fragmentation in less-developed areas. To mitigate these issues, policymakers could leverage existing agricultural cooperatives as distribution nodes and promote public–private partnerships to streamline supply chain management. Additionally, regulatory bodies should establish collaborative frameworks with certification agencies and digital platforms to ensure that eco-label verification processes are transparent, standardized, and accessible to both producers and consumers.

To address these constraints, policymakers should prioritize targeted, actionable policy instruments that reduce consumers’ perceived behavioral costs. For instance, local governments could pilot community-based green food voucher programs or embed green food subsidies within existing public food assistance schemes, particularly for vulnerable and low-income groups. In under-resourced or rural regions, the establishment of government-supported regional distribution platforms may enhance the physical availability of certified green food. Furthermore, school-based intervention strategies could be institutionalized by integrating green food education into environmental and health curricula, alongside the promotion of “green school cafeterias” that prioritize eco-certified procurement practices.

Simultaneously, enhancing institutional transparency and strengthening the credibility of certification standards remains paramount. Digital technologies such as blockchain-enabled traceability systems, QR-code verification mechanisms, and real-time platform-based certification visibility can substantially elevate consumer trust in eco-labels and reduce informational asymmetries. Nevertheless, the technological implementation of traceability systems must overcome consumers’ digital literacy gaps, particularly in rural regions where QR-code scanning and blockchain-based platforms may be underutilized. To address this, simplified verification tools—such as color-coded visual indicators or on-site certification displays—can complement digital mechanisms, ensuring inclusivity. Policymakers should also consider launching public education campaigns that demystify these technological processes, enhancing consumer confidence and usability across demographic strata.

However, technological innovation alone is insufficient. Regulatory agencies must also consolidate and reform the current eco-labeling system, harmonizing definitions and visual standards across “green,” “organic,” and “pollution-free” categories. The implementation of tiered labeling systems (e.g., Grade A/B/C) and mandatory QR-code integration into packaging can improve consumer interpretability, mitigate skepticism, and facilitate informed purchase decisions.

From a market engagement perspective, the retail ecosystem—including both e-commerce platforms and physical retail channels—should transition toward experience-driven communication strategies. Given that green food increasingly represents a symbolic lifestyle choice rather than merely a health-driven product, affective engagement should be amplified across consumer touchpoints. Practices such as personalized content curation, peer-influenced sharing mechanisms, and immersive digital storytelling can help embed green food into consumers’ daily routines, fostering not only behavioral continuity but also value-based identity alignment. However, transitioning to experience-driven engagement requires retailers to overcome the challenge of translating abstract sustainability narratives into tangible consumer experiences. This necessitates innovative approaches such as in-store eco-experience zones, interactive online campaigns featuring user-generated green stories, and gamified reward systems that incentivize sustainable behaviors. For e-commerce platforms, integrating green food choices into daily lifestyle content—such as recipe sharing, influencer collaborations, and personalized green living tips—can effectively bridge the intention–behavior gap.

Overall, the results show that the green food communication strategy needs to be reconstructed urgently, and the affective dimensions and socially situated cues should be organically integrated into the public communication framework. Consumers’ decision logic is no longer driven solely by information or price, but is deeply embedded in a network of relationships consisting of trust, emotion, lifestyle aspiration, and peer influence. This requires communication practices to shift from a persuasion model based on rational logic to a path that emphasizes shared meanings and collective identity narratives. In this process, public education should focus on strengthening experiential storytelling, presenting representative green role models, and framing green choices as a normative, emotionally fulfilling, and socially endorsed behavior. Public education should not only emphasize emotional storytelling and green role models but also be institutionalized through curriculum integration in primary and secondary schools, training modules for educators, and community-based environmental literacy programs. Government agencies and NGOs could co-develop educational toolkits that align green food awareness with broader sustainability goals. Only when product development, policy direction, and media discourse can be deeply connected with consumers’ emotional expectations and identity construction can green food truly achieve the transformation from a policy goal to a culturally embedded and behaviorally sustained consumption practice.

Ultimately, the effective transformation of green food consumption requires a synchronized effort across multiple stakeholder levels. Product development, policy instruments, media discourse, and consumer engagement strategies must coalesce into a cohesive ecosystem that aligns functional credibility with emotional resonance. Without this integrated approach, isolated interventions risk being undermined by structural inconsistencies and fragmented messaging, thereby limiting their capacity to foster enduring behavioral change.

## 6. Conclusions

This study investigated the formation of green food purchase intention by conceptualizing the dynamic interaction between internal psychological dispositions and external situational factors. The findings reveal that external influences—such as flow experience, health beliefs, social influence, price perception, label ambiguity, access limitations, and behavioral resistance—indirectly shape consumers’ purchasing intentions through three core internal mediators: green trust, perceived value, and lifestyle congruence. This cognitive–affective negotiation mechanism underscores that green food purchase intention is not a product of linear rational evaluation, but rather an intricate synthesis of cognitive assessments and affective–symbolic interpretations embedded within personal beliefs and socio-cultural contexts.

The theoretical model developed in this study advances existing paradigms by elucidating how external cues (e.g., structural constraints, social referencing) interact with internal cognitive–emotional processes (e.g., trust-building, value alignment, identity signaling) to facilitate or inhibit intention formation. By integrating both situational stimuli and emotional mediators, this framework offers a multi-layered and context-sensitive perspective on sustainable consumption behavior, enriching the explanatory power of traditional models such as the TPB and HBM.

While the study provides valuable insights, it also acknowledges several limitations that constrain its empirical generalizability. The sample is confined to middle-income urban consumers in contemporary China—a demographic characterized by greater access to green food channels and heightened eco-literacy. Consequently, the behavioral mechanisms identified, particularly pathways of emotional trust and identity alignment, are deeply embedded in China’s socio-cultural context, where state-endorsed certifications and collective lifestyle narratives play a pivotal role in shaping consumer trust structures. These pathways may manifest differently in regions with alternative socio-political trust ecosystems or varying degrees of eco-labeling credibility. Therefore, extrapolating the current findings to marginalized, rural, or structurally constrained populations necessitates cautious interpretation and further empirical validation.

Moreover, the absence of large-scale statistical validation limits the model’s immediate predictive applicability across broader consumer segments. However, given the exploratory aim of this research, the adoption of the grounded theory methodology is particularly justified. This study’s objective was not to establish statistically generalizable patterns, but rather to excavate latent emotional structures and behavioral reasoning mechanisms that are often overlooked in pre-defined, variable-driven models. Within this interpretive paradigm, the depth, coherence, and conceptual rigor of the emergent model—especially the cognitive–affective negotiation framework—demonstrate strong theoretical transferability to contexts where consumer decisions are similarly shaped by identity, trust, and symbolic alignment.

Nonetheless, to validate and refine this model’s robustness, future studies should incorporate large-sample quantitative approaches, such as Structural Equation Modeling (SEM), alongside cross-cultural comparative analyses. Particularly, research in Western or emerging markets with differing institutional trust dynamics will be essential to examine how emotional trust, value resonance, and identity pathways adapt or recalibrate in varied cultural milieus. Additionally, expanding the participant pool to encompass underrepresented groups—such as rural residents, low-income consumers, or structurally marginalized cohorts—would provide critical insights into the heterogeneity of sustainable consumption behaviors.

This study also highlights the necessity of mixed-methods research designs that integrate qualitative narrative analysis with quantitative validation techniques. Such methodological triangulation would enable a more comprehensive understanding of the cognitive–affective dynamics underlying green food purchase intention, offering empirical precision while retaining contextual sensitivity.

In sum, while grounded in the Chinese socio-cultural landscape, this study’s theoretical contributions are not confined to it. By foregrounding the synergistic interplay between emotional mediators and structural constraints, the proposed model offers a generative conceptual architecture for future explorations of sustainable consumption. It opens fertile avenues for multi-disciplinary, cross-cultural, and methodologically pluralistic research that seeks to unravel the socio-emotional intricacies of ecological consumer behavior.

## Figures and Tables

**Figure 1 foods-14-02856-f001:**
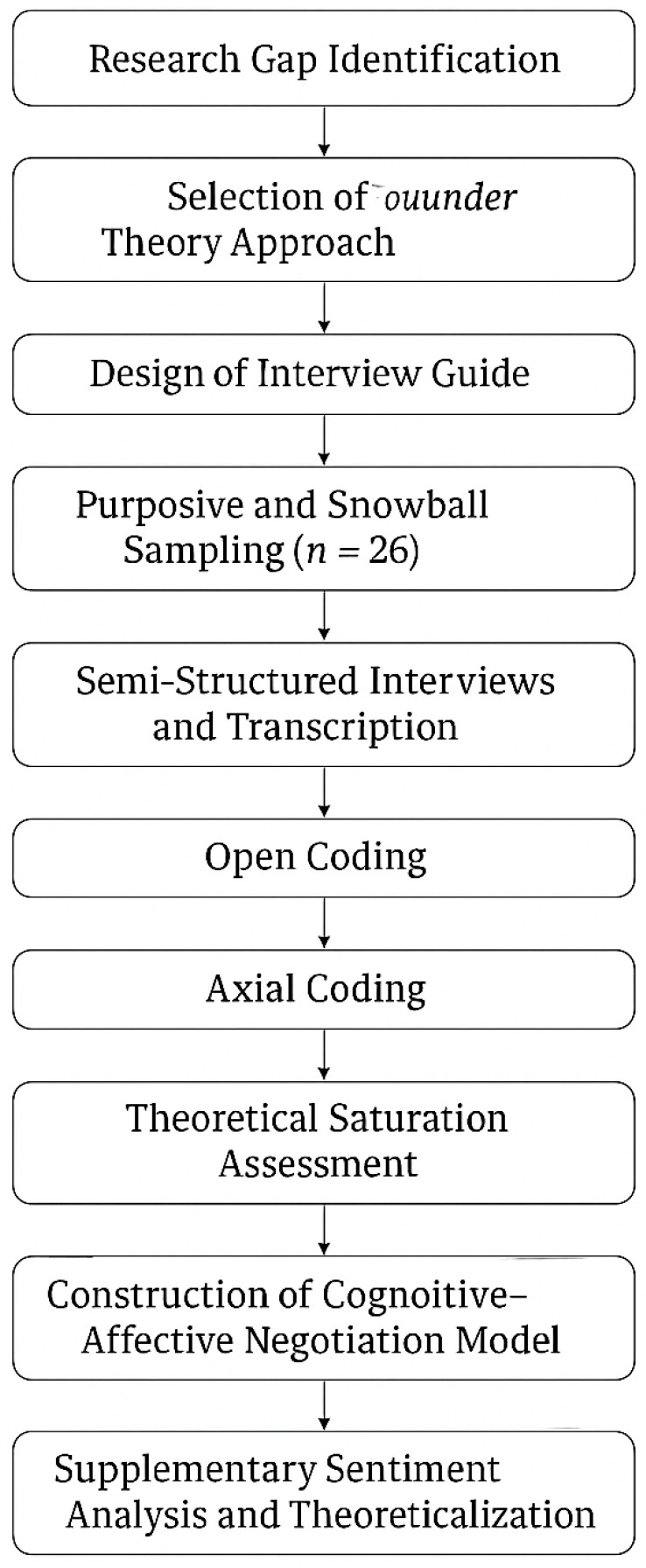
Research process based on grounded theory for exploring green food purchase intention.

**Figure 2 foods-14-02856-f002:**
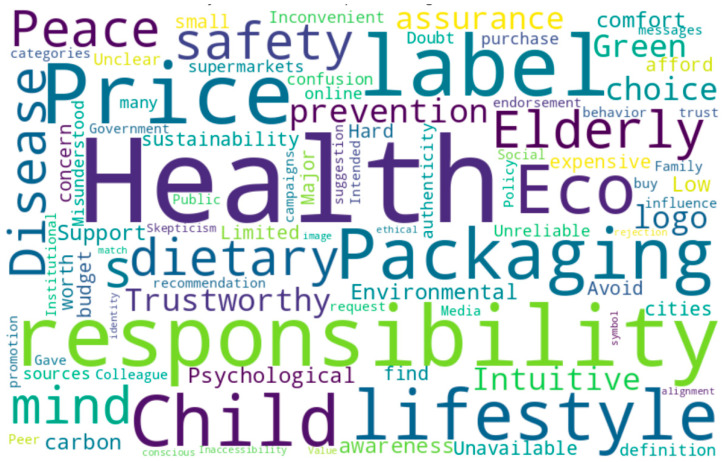
Keyword word cloud map.

**Figure 3 foods-14-02856-f003:**
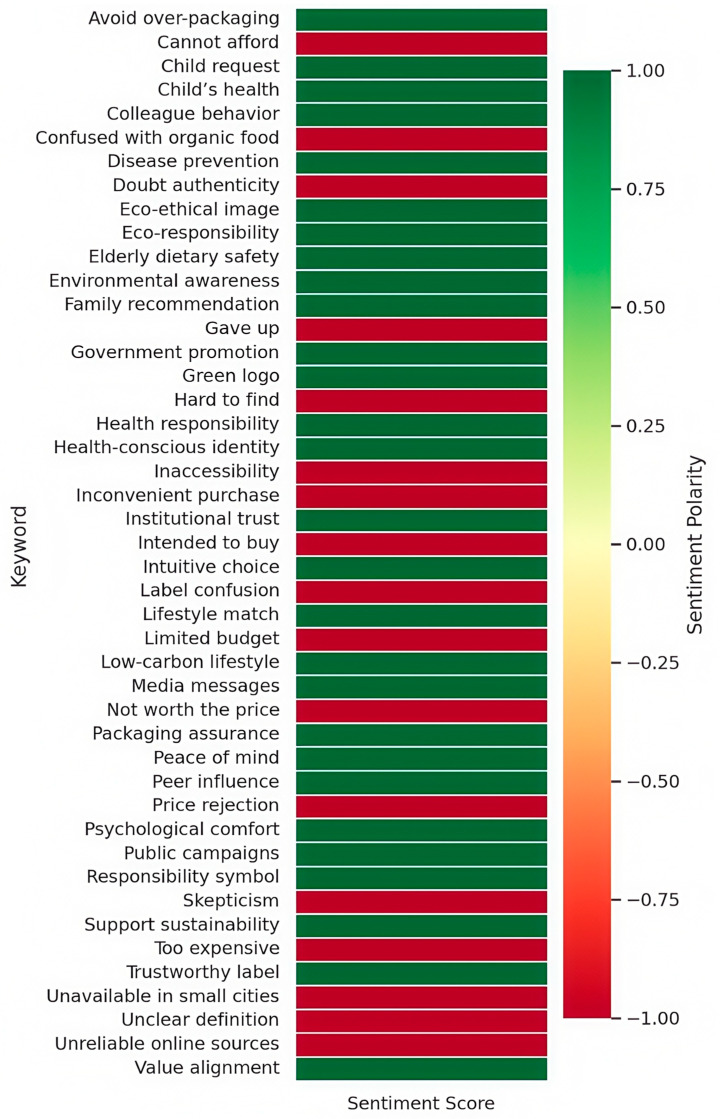
Sentimental polarity heatmap of open coding keywords.

**Figure 4 foods-14-02856-f004:**
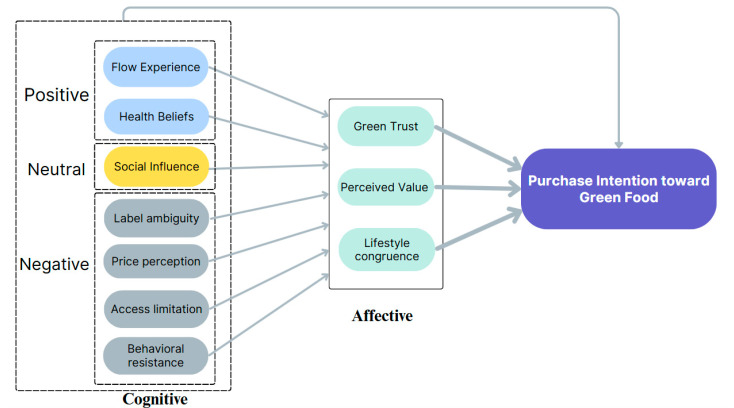
The cognitive–affective negotiation of green food consumption intention.

**Table 1 foods-14-02856-t001:** Factors influencing green food purchase intention.

Factor	Methodology	Findings	Research Context	Source
Environmental awareness, health consciousness, perceived value, emotional identification	Conceptual synthesis	These internal factors shape ecological attitudes, perceived benefits, and emotional resonance, forming the basis of green food intention	China, TPB-based studies	Li and Shan [17], Jiang, Lau [18]
Pricing, product availability, labeling trust, peer influence, institutional credibility	Mixed methods (survey, SEM)	External constraints such as price sensitivity and label trust significantly affect behavioral feasibility and perceived credibility	Cross-cultural, policy-sensitive contexts	Sheikh, Tourani [19], Siuda and Grębosz-Krawczyk [20], Yasin, Shalifullizam [21]
Consumer knowledge and attitude	Survey + regression	Higher consumer knowledge and more positive attitudes significantly enhance purchase intention	India, consumers aged 20+	Ganesh, Mohamed [22]
Emotional experience (feeling of being loved, identity confirmation)	Survey (emotion scale + intention measure)	Emotional attachment enhances green identity and increases willingness to adopt green food behavior	Southeast Asia, symbolic consumption	Tang, Chen [23]
Social media interaction (interactive brand engagement)	SEM (digital marketing context)	Entertainment value, social connection, and informative interaction enhance green food purchase motivation	Japan, digital platform users	Hiroshima and Nismi [24]
Eco-friendly packaging and design	Experimental design + SEM	Eco-packaging improves perceived brand sustainability and triggers environmental purchase intentions	Poland, in-store consumer studies	Siuda and Grębosz-Krawczyk [20]
Trust in green labels, claims, institutions	Survey + SEM	Institutional and certification trust outweigh product-specific traits in motivating green purchasing	Global studies on trust	Sheikh, Tourani [19]
Transparent communication strategies	Field experiments + SEM	Transparent messaging reduces skepticism, increases credibility, and mitigates greenwashing effects	European green marketing campaigns	Tetrevova, Kotkova Striteska [25]
Environmental concern, perceived behavioral control, subjective norms	TPB-based survey model	These TPB constructs effectively predict behavioral intention toward green food	China, urban consumers	Jiang, Lau [18]
Health/environmental values in TPB extensions	Extended TPB + SEM	Strong health beliefs and ecological values amplify attitude and intention toward organic/green food	China, mid-income populations	Li and Shan [17]
Gen-Z distrust due to greenwashing and COVID-related uncertainty	Survey + regression	Gen-Z holds strong green values, but perceived inauthenticity reduces their intention to buy	China, post-pandemic youth	Zhang, Quoquab [26]
Peer influence, health perception, product attitude	Survey + PLS-SEM	Social norms, perceived health benefits, and positive attitudes drive organic food purchase among youth	Malaysia, young adult cohort	Yasin, Shalifullizam [21]
Past behavior, green knowledge, trust	Survey + TPB extension model	Past behavior and institutional trust significantly predicted green food purchase intentions	Poland, individual consumers	Witek and Kuźniar [27]
Green perceived value, trust, satisfaction	SEM, cross-country comparison	Perceived product quality and price–value ratio influence satisfaction and purchase intention	Germany and Brazil, middle-income consumers	Graça and Kharé [28]
Price sensitivity, environmental concern	Survey + mediation analysis	Price awareness mediates the relationship between environmental concern and green purchase intention	Portugal, general consumers	Lopes, Silva [29]
Label credibility, institutional trust	Survey + SEM	Institutional trust in eco-labels significantly predicts purchase intentions for environmentally friendly meat	Italy, environmentally conscious consumers	Stranieri, Ricci [30]
Sustainable intentions, ecological values	TPB-based survey	Attitude, perceived behavioral control, and environmental concern predict green entrepreneurial and purchase intentions	Colombia, university students	Romero-Colmenares and Reyes-Rodríguez [31]
Organic food trust, eco-motivation	Multi-national survey	Eco-motivation and trust in organic labels are strong predictors of purchase intention across regions	Oceania and Europe, cross-cultural sample	Jaeger, Chheang [32]
Attitudes, habits, lifestyles	SEM and necessary condition analysis (NCA)	Identifies habitual behaviors and lifestyle congruence as critical, yet underexplored, enablers of green food purchasing; highlights the interaction between routine practices and ecological attitudes	Multi-country survey, cross-cultural validation	Mazhar and Zilahy [14]
Perceived usefulness, environmental problem awareness, pandemic fear	Survey regression	Demonstrates that fear of pandemic recurrence and environmental problem awareness significantly affect young consumers’ purchase intentions, beyond traditional rational predictors	China, Gen-Z post-pandemic cohort	Zhang, Quoquab [15]
Individual characteristics, cognitive degree	Survey regression	Establishes that personal cognitive capacity and demographic traits substantially influence green food consumption behavior, but lacks exploration of affective pathways	China, general consumers	Deng, Huang [13]

**Table 2 foods-14-02856-t002:** Demographic composition and theoretical relevance of interview participants.

Participant Group	Age Range	Occupation/Role	Living Area	Sample Size (n)	Rationale for Inclusion	Theoretical Relevance
Parents or caregivers of children/elders	30–55	Educators, freelancers, service workers	Urban areas	6	High sensitivity to food safety and family health concerns	Related to health beliefs and internal motivation
University students and early-career professionals	18–29	Students, white-collar employees	Urban areas	8	Exposed to environmental discourses, peer influence, and identity branding	Related to identity motivation and social influence
Professionals in health, education, and agriculture	28–50	Doctors, teachers, agricultural technicians	Semi-urban and mixed areas	5	Professionally exposed to ecological responsibility and institutional standards	Related to institutional trust and ecological values
Rural and small-town consumers	35–65	Farmers, small vendors, informal workers	Rural or third-tier cities	7	Confront practical constraints in price, access, and label comprehension	Related to structural barriers and external constraints

**Table 3 foods-14-02856-t003:** Theoretical foundations of the semi-structured interview guide.

Interview Section	Key Constructs Covered	Theoretical Foundations	Relevant Theoretical Concepts/Variables
I. Background Information	Demographic context, prior experience	Grounded Theory (Glaser and Strauss)	Ensures diversity, supports theoretical sampling and saturation
II. Perception and Initial Attitude	General attitudes, baseline awareness	Theory of Planned Behavior (TPB); Health Belief Model (HBM)	Attitude toward behavior, perceived susceptibility, perceived benefits
III. Internal Influencing Factors	Health motivations, environmental values, emotional responses, identity	Health Belief Model (HBM); Value–Belief–Norm Theory (VBN); Identity Theory	Health consciousness, environmental concern, personal norms, symbolic identity
IV. External Influencing Factors	Price, accessibility, social and media influence, label trust	TPB; Institutional Trust Theory; Social Influence Theory	Subjective norms, perceived behavioral control, institutional credibility
V. Intention–Behavior Gap and Constraints	Behavior inhibition, conversion barriers	Constraint Theory; Intention–Behavior Gap Literature	Structural limitations, decision ambivalence, perceived behavioral feasibility
VI. Closing and Open-Ended Questions	Emergent themes and participant-led insight	Grounded Theory (Theoretical Sensitivity)	Allows discovery of new categories beyond pre-existing models

**Table 5 foods-14-02856-t005:** Axial coding results.

Axial Theme	Associated Open Coding Categories	Representative Keywords/Expressions
Health Motivation	Health Motivation	“Child’s health”, “Elderly dietary safety”, “Peace of mind”, “Disease prevention”
Environmental Values	Environmental Attitude	“Eco-responsibility”, “Support for sustainability”, “Low-carbon lifestyle”, “Doing something for the environment”
Institutional Trust	Emotional Trust	“Trustworthy label”, “Intuitive choice”, “Psychological comfort”, “Green logo means safety”
Label Uncertainty	Label Confusion	“Too many labels”, “Unclear what green food is”, “Confused with organic food”, “Label misunderstanding”
Perceived Value Judgment	Price Concern	“Too expensive”, “Not worth the price”, “Limited budget”, “High value, low accessibility”
Experiential Immersion	Flow Experience (derived from open coding under Positive Emotional Involvement)	“Enjoyable purchase process”, “Immersive experience”, “Platform-driven satisfaction”, “Control in choice”
Social Referencing	Social Influence	“Recommended by family”, “Tried it because friends did”, “Children requested it”, “Peer encouragement”
Policy-Induced Cognition	Policy and Media Messaging	“Government campaign”, “Media says it is healthier”, “Public messaging”, “Expert recommendations”
Availability Constraints	Access and Availability	“Hard to find”, “Only in major supermarkets”, “Not available in small cities”, “Inconvenient to purchase”
Behavioral Resistance	Purchase Barriers	“Intended to buy but gave up”, “Uncertainty about authenticity”, “Skepticism”, “Distrust in online sources”

**Table 6 foods-14-02856-t006:** Key variables identified in the selective coding framework.

Variable	Definition
Flow experience	A psychologically immersive and emotionally engaging state during the green food purchasing process that enhances satisfaction and behavioral continuity.
Health beliefs	Consumers’ convictions about the health benefits of green food, including perceptions of nutritional safety, disease prevention, and family well-being.
Green trust	The integrated belief in the authenticity, reliability, and emotional safety of green food, derived from both institutional credibility and affective associations.
Perceived value	The consumer’s comprehensive evaluation of green food’s functional utility, ethical significance, and symbolic resonance relative to its cost.
Price perception	The extent to which green food is considered financially burdensome, limiting purchase behavior even when attitudes are positive.
Label ambiguity	Consumer confusion arising from unclear, overlapping, or misinterpreted labeling systems that hinder confidence in product authenticity.
Social influence	The shaping of green food purchasing intention through perceived expectations or behavioral cues from peers, family, or close social networks.
Access limitation	Structural barriers such as product unavailability, restricted distribution, or inconvenient access that hinder consumers’ purchasing behavior.
Behavioral resistance	Psychological or contextual obstacles—such as skepticism, effort cost, or transaction friction—that inhibit the transition from intention to action.
Lifestyle congruence	The degree to which green food consumption resonates with personal identity, ethical self-concept, and everyday lifestyle patterns.

## Data Availability

No new data were created or analyzed in this study. Data sharing is not applicable to this article.

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
