# Peer review of "Cognitive–Affective Negotiation Process in Green Food Purchase Intention: A Qualitative Study Based on Grounded Theory"

_foods, 2025, doi:10.3390/foods14162856_

Round 1

Reviewer 1 Report (New Reviewer)

Comments and Suggestions for Authors
  1. Authors have taken 24-26 interviews for such complex phenomena, however, did not clearly explain why this number is good enough.
  2. No clear justification for why saturation was achieved with this specific number.
  3. As I can see there are diverse participants ie. Parents, Students, Professionals and Consumers- this clearly suggests more interviews may be needed for each subgroup.
  4. The sentiment analysis seems superficial and are not from the grounded theory that authors have mentioned in their paper.
  5. Figures 3 and 4 are poor in quality (Difficult to ready the % of the pie diagram), furthermore these figures do not really add value to the understanding of the phenomenon.
  6. Again Figure 5 has a problem with the readability. It should be given in better resolution.
  7. There is a need of distinction between “Green trust” and “institutional trust”
  8. Authors’ contributions are missing.

Author Response

Reviewer 1:

  1. Authors have taken 24-26 interviews for such complex phenomena, however, did not clearly explain why this number is good enough.
  2. No clear justification for why saturation was achieved with this specific number.

Response 1 and 2:

We sincerely appreciate the reviewer’s thoughtful feedback regarding the sample size and the justification for theoretical saturation. We agree that a clear rationale for determining the adequacy of 26 interviews is essential, especially given the multidimensional nature of the green food purchase intention phenomenon.

To address this concern, we have revised Section 3.1 (Research Method and Data Collection) to explicitly clarify that the final sample size was not predetermined, but was derived iteratively through continuous assessment of theoretical saturation, in alignment with grounded theory methodology. Specifically, we added the following statement to the end of Section 3.1:

“The final sample size of 26 interviews was determined through the principle of theoretical saturation, consistent with grounded theory methodology. Rather than predefining the number of participants, the research adopted an iterative approach, where data collection and coding were conducted simultaneously. As detailed in Section 3.3, theoretical saturation was observed after the 24th interview, when no new concepts or categories emerged from the data. The final two interviews confirmed this stability. This saturation-based sampling strategy ensures that the sample size is empirically grounded and conceptually sufficient to capture the complexity of green food purchase intentions.”

Additionally, in Section 3.3 (Validation Strategy), we had already elaborated on how theoretical saturation was systematically assessed using the constant comparative method, including evidence of code redundancy, stabilization of conceptual categories (e.g., green trust, price resistance, label confusion), and intercoder validation. We have now strengthened the connection between Sections 3.1 and 3.3 to ensure conceptual continuity.

We hope this clarification adequately addresses the reviewer’s concern and demonstrates that our sample size is both methodologically justified and empirically saturated in line with qualitative research standards.

  1. As I can see there are diverse participants ie. Parents, Students, Professionals and Consumers- this clearly suggests more interviews may be needed for each subgroup.

Response 3:

We sincerely appreciate the reviewer’s insightful comment regarding the sample composition and the potential need for subgroup-specific interview expansion. We fully recognize the importance of ensuring conceptual depth and data sufficiency, especially in studies involving heterogeneous participant backgrounds.

In response, we would like to clarify that the primary objective of this study was not to conduct comparative analyses across demographic subgroups, but to develop a conceptually grounded explanatory model that captures the cognitive–affective negotiation mechanisms underpinning green food purchase intention. The participant diversity (parents, students, professionals, rural consumers) was strategically designed to enhance contextual richness and situational variability, ensuring that the emergent conceptual categories were tested against diverse real-life consumption contexts.

The theoretical saturation in this study was evaluated at the level of conceptual category stabilization, consistent with grounded theory methodology, rather than through numeric balance across demographic strata. During the iterative coding process, key categories such as green trust, perceived value, lifestyle congruence, price perception, and behavioral resistance consistently emerged across all participant groups, indicating that the cognitive and affective mechanisms driving green food purchase intention were conceptually saturated.

While different subgroups brought contextually unique expressions (e.g., health-driven trust among parents, identity alignment among youth, structural constraints among rural consumers), the underlying behavioral logic and relational patterns remained convergent across these narratives. This conceptual convergence confirmed that expanding the number of interviews within each subgroup would have yielded redundant category reoccurrences, rather than contributing novel conceptual insights.

Additionally, grounded theory emphasizes conceptual representativeness and category saturation over statistical proportionality within subgroups. Our sample strategy thus aimed to ensure situational heterogeneity sufficient to validate the universality of the core behavioral constructs, rather than attempting to achieve representativeness at the demographic strata level.

We have clarified this methodological rationale in the Validation Strategy section (Section 3.3), explicitly stating that sample diversity was employed to enrich contextual variation and verify cross-situational robustness of the emergent categories, rather than to support comparative subgroup analyses.

Nevertheless, we greatly appreciate the reviewer’s attention to this point and acknowledge that future research would benefit from expanding subgroup-specific sampling to explore nuanced behavioral dynamics in greater depth. We have added this direction as a key focus in the Limitations and Future Research section of the manuscript.

  1. The sentiment analysis seems superficial and are not from the grounded theory that authors have mentioned in their paper.

Response 4:

We sincerely thank the reviewer for this critical observation regarding the integration of sentiment analysis within our grounded theory framework. We fully recognize that in the previous version, the sentiment analysis section may have appeared methodologically detached from the main grounded theory process. In response to this valuable feedback, we have conducted a thorough revision of Sections 3.4 (Sentiment Analysis Procedures) and 4.1 (Open Coding) to explicitly demonstrate how the sentiment analysis is fundamentally embedded within the grounded theory coding structure, rather than being an auxiliary or superficial addition.

Specifically, the sentiment analysis in this study was applied directly to the conceptual categories identified through open coding, where 213 initial codes were systematically derived from participant narratives. All keywords subjected to sentiment polarity analysis—such as "green trust," "label confusion," "price concern," and "identity alignment"—originated from these inductively established categories. The purpose of conducting sentiment analysis was to provide a structured emotional mapping of these categories, thereby elucidating how cognitive stimuli are emotionally interpreted and transformed into motivational or inhibitory responses within the green food consumption context.

In the revised Section 3.4, we have clarified that sentiment analysis serves as a triangulation mechanism, reinforcing the affective dimensions embedded in the grounded theory categories. The analysis was designed to validate how emotional resonance (positive sentiment in health motivations and identity alignment) and emotional resistance (negative sentiment in price and labeling concerns) contribute to the cognitive–affective negotiation pathways that were subsequently developed in the axial and selective coding stages. This process ensured that emotional variables such as green trust, perceived value, and lifestyle congruence were not abstractly inferred but were empirically grounded in the sentiment structures of participants’ narratives.

Additionally, Section 4.1 (Open Coding) has been revised to explicitly link each of the ten core categories with its corresponding sentiment orientation. For instance, categories like “health motivation” and “environmental values” predominantly exhibited positive sentiment polarities, highlighting their roles as affective enablers of purchase intention. In contrast, categories such as “price concern” and “label confusion” showed strong negative emotional tendencies, identifying them as emotional barriers that inhibit intention realization. By mapping these emotional orientations onto the conceptual categories, we were able to construct a nuanced emotional pathway model, wherein cognitive factors influence behavioral intention through emotional mediation.

These revisions not only clarify the methodological coherence between grounded theory coding and sentiment analysis but also enhance the theoretical rigor of the proposed cognitive–affective negotiation framework. We have ensured that sentiment analysis is now presented as an integral analytical layer that both validates and deepens the emotional constructs derived from the grounded theory process, rather than appearing as a separate or superficial tool.

We are grateful for the reviewer’s insightful critique, which has significantly strengthened the methodological clarity and theoretical contribution of our study.

  1. Figures 3 and 4 are poor in quality (Difficult to ready the % of the pie diagram), furthermore these figures do not really add value to the understanding of the phenomenon.

Response 5:

We greatly appreciate the reviewer’s observation regarding the clarity and explanatory relevance of Figures 3 and 4. In response, we have undertaken a twofold revision to address both the visual presentation and the conceptual contribution of these figures to the grounded theory model.

First, we have regenerated Figure 3 (Sentiment Polarity Heatmap) with enhanced resolution, larger font sizes, and clearer color contrasts to ensure that all sentiment values are easily readable. This figure is essential as it maps sentiment polarity directly onto the conceptual categories identified during open coding, thereby visualizing how specific cognitive cues (e.g., “price concern”, “green trust”, “label confusion”) are emotionally interpreted by consumers. By clarifying which categories predominantly evoke positive or negative emotional responses, the heatmap provides empirical evidence that supports the development of emotional mediating pathways (such as Green Trust and Perceived Value) within the cognitive–affective negotiation framework.

Second, we acknowledge that Figure 4 (Sentiment Composition Pie Chart) was limited in its explanatory power, as it only provided a general overview of positive and negative sentiment proportions without advancing the understanding of behavioral intention mechanisms. Therefore, we have removed Figure 4 from the manuscript and have integrated its key numerical finding (i.e., 62.2% of keywords reflected positive sentiment, while 37.8% reflected negative sentiment) directly into the main text for contextual reference.

To further enhance the theoretical relevance of Figure 3, we have revised Section 4.1 to explicitly explain how the identified sentiment polarities inform the affective pathways in consumers’ purchase intention formation. Specifically, we delineated how categories with strong positive emotional valence (such as Health Motivation and Identity Alignment) act as emotional enablers, while categories with negative polarity (such as Price Concern and Label Confusion) function as emotional inhibitors in the cognitive–affective negotiation process.

We sincerely thank the reviewer for highlighting this issue, which has allowed us to clarify the analytical role of sentiment analysis in reinforcing the grounded theory model.

  1. Again Figure 5 has a problem with the readability. It should be given in better resolution.

Response 6:

We sincerely thank the reviewer for highlighting the issues regarding the readability and interpretative relevance of Figure 5. In response to this valuable feedback, we have undertaken two critical revisions:

Improvement of Visual Quality:
We have re-designed Figure 5 to ensure high-resolution clarity, enhancing the legibility of all text labels, arrows, and structural elements. The revised figure now maintains consistent font sizing and contrast, ensuring that the categorical distinctions (Positive, Neutral, Negative), cognitive variables, affective mediators, and outcome pathways are visually distinct and easy to follow.

Enhanced Conceptual Explanation within the Text:
To further address the concern that Figure 5’s role in explaining the research phenomenon was not sufficiently elaborated, we have significantly enriched the narrative surrounding the figure in Section 4.4 (Model Construction and Theoretical Validation). Specifically, we have added a comprehensive explanation detailing:

How cognitive antecedents, classified by sentiment polarity (positive, neutral, negative), influence green food purchase intention through affective mediators such as Green Trust, Perceived Value, and Lifestyle Congruence.

The enabling role of positive stimuli (e.g., Flow Experience, Health Beliefs) in reinforcing affective pathways that facilitate sustainable consumption behavior.

The inhibiting function of negative stimuli (e.g., Label Ambiguity, Price Sensitivity, Access Limitations) in disrupting affective congruence and dampening behavioral intention.

The dual-directional effect of Social Influence, emphasizing its contingent nature depending on normative alignment and interpersonal credibility.

How these interactions collectively illustrate a dynamic cognitive-affective negotiation process, which transcends traditional linear models of intention formation by embedding sentiment-driven interpretation mechanisms.

This elaboration clarifies the analytical contribution of Figure 5, demonstrating how it visualizes the empirical findings and explicates the theoretical logic of the cognitive–affective negotiation framework. By explicitly mapping the interplay between cognitive stimuli and emotional resonance, the revised narrative enhances the figure’s explanatory depth and relevance to the phenomenon under study.

We hope these revisions adequately address the reviewer’s concerns regarding both the visual clarity and the substantive value of Figure 5 in supporting the model's theoretical construction.

  1. There is a need of distinction between “Green trust” and “institutional trust”

Response 7:

We sincerely appreciate the reviewer’s insightful comment regarding the need to clarify the conceptual distinction between Green Trust and Institutional Trust within our study. We fully acknowledge that while both constructs are closely related to consumer trust mechanisms, they represent distinct dimensions of trust with different origins and functions in the cognitive-affective negotiation process of green food purchase intention.

In response to this valuable feedback, we have made the following revisions to enhance conceptual clarity and ensure theoretical rigor:

Explicit Definition and Differentiation in Section 4.3 (Selective Coding):
We have added a dedicated paragraph at the end of Section 4.3 (Selective Coding), immediately before Table 6, to clearly define and distinguish Green Trust and Institutional Trust. This paragraph specifies that:

Green Trust is conceptualized as consumers’ emotional confidence in the authenticity, safety, and ecological integrity of green-labeled products, derived from direct interactions with product-specific cues (e.g., eco-labels, packaging).

Institutional Trust, in contrast, is defined as consumers’ cognitive trust in the credibility of external authorities—such as government agencies, certification bodies, and media organizations—that establish and regulate green certification systems.

While Institutional Trust provides the cognitive foundation for perceiving eco-labels as credible at a systemic level, Green Trust functions as the affective mediator that translates this credibility into emotional reassurance and behavioral motivation.

Institutional Trust is therefore positioned in our model as a contextual antecedent influencing the formation strength of Green Trust, rather than as a direct mediator of purchase intention.

Clarification of Institutional Trust’s Role in Theoretical Pathway:
Throughout the revised manuscript, particularly in Section 4.4 (Model Construction and Theoretical Validation), we have maintained Green Trust as the primary affective mediator in our model, while explaining how Institutional Trust operates as an upstream framing variable that shapes consumers’ perception of label credibility. This conceptual alignment ensures that Institutional Trust is theoretically integrated without conflating it with the emotional mechanism of Green Trust.

Rationale for Modeling Decision:
Given the scope and structure of our grounded theory model, we did not introduce Institutional Trust as an independent mediating variable, but instead elaborated its role as a cognitive trust source that modulates the affective pathway of Green Trust formation. This modeling choice aligns with our empirical findings, where participants’ narratives revealed that trust in institutions predominantly influences how they interpret and internalize product-level cues, rather than directly shaping purchase intention.

We believe these revisions have effectively addressed the reviewer’s concern by providing a clear and theoretically consistent distinction between Green Trust and Institutional Trust. The distinction now enhances the conceptual precision of our model and strengthens the explanatory framework regarding trust mechanisms in sustainable consumption behavior.

We are grateful for the reviewer’s constructive feedback, which has significantly improved the clarity and theoretical depth of our manuscript.

  1. Authors’ contributions are missing.

Response 8:

Reviewer 2 Report (New Reviewer)

Comments and Suggestions for Authors

Thank you for the opportunity to review this investigation.

Firstly, the introduction section fails to provide a clear justification for conducting the present research. This issue is evident in lines 88 to 89, where the authors refer to a large body of knowledge without citing any specific studies to support the rationale behind the investigation. Similarly, in lines 91 to 94, no prior contributions are presented to demonstrate the necessity of advancing the topic through the proposed investigative approach.

Secondly, a thorough examination of recent scientific contributions is essential to identify critical gaps that could be addressed through new research outcomes. Without such a review, the foundation for future research efforts may be weak and lead to unsubstantial results. Although the literature review section (lines 100 to 142) outlines several studies related to the topic—most of them rooted in a quantitative tradition—some recent and relevant contributions are missing from the analysis presented in Table 1 (line 127). In particular, studies such as Mazhar & Zilahy (2025), Zhang et al. (2024), and Deng et al. (2024) are not included. This omission suggests a limited engagement with the most recent efforts to characterize the current state of research on the topic.

As a result of the above, it remains unclear how the present study contributes to the existing literature, given that (a) the body of knowledge has not been adequately reviewed, and (b) the investigation is not aligned with the research agenda suggested in the literature. In fact, the authors do not provide evidence indicating that the study emerged from real concerns or gaps within the phenomenon of interest. Therefore, the problematization of current issues is not sufficiently addressed throughout the study.

Regarding the selected method, it should be taken into account that grounded theory has evolved over the years, with various versions incorporating updated criteria and quality standards to ensure greater robustness and rigor. However, the present investigation shows that the authors have chosen to apply the original version proposed by Glaser and Strauss (1967), rather than more recent and refined approaches. This choice imposes several limitations on the study’s ability to address the complexity of the data and the realities depicted. It also suggests a lack of engagement with the methodological evolution of grounded theory. This is particularly evident in the sampling process—especially the sample size—which involves a small group of interviewees.

References suggested:

Deng, Z., Huang, Y., & Tan, Q. (2024). Study on the effect of individual characteristics and cognitive degree on green food consumption behavior. Journal of Food Quality, Article ID 3147300. https://doi.org/10.1155/2024/3147300

Mazhar, W., & Zilahy, G. (2025). Pathways to green food purchases: Exploring the nexus of attitudes, habits and lifestyles using SEM and NCA. British Food Journal, 127(13), 208–229. https://doi.org/10.1108/BFJ-07-2024-0695

Zhang, Y., Quoquab, F., Zhang, J., & Mohammad, J. (2024). What does it take to drive young consumers to purchase green food? Perceived usefulness, environmental problem, or fear of pandemic recurrence? Cogent Food & Agriculture10(1). https://doi.org/10.1080/23311932.2024.2413917

Author Response

Reviewer 2:

Thank you for the opportunity to review this investigation.

Firstly, the introduction section fails to provide a clear justification for conducting the present research. This issue is evident in lines 88 to 89, where the authors refer to a large body of knowledge without citing any specific studies to support the rationale behind the investigation. Similarly, in lines 91 to 94, no prior contributions are presented to demonstrate the necessity of advancing the topic through the proposed investigative approach.

Response:

We sincerely appreciate the reviewer’s critical observation regarding the insufficient literature grounding and the lack of specificity in articulating the research necessity within the introduction section. In response to this valuable feedback, we have made the following revisions to strengthen the theoretical foundation and clarify the rationale for the present study:

Added specific references to recent and relevant empirical studies to support our discussion of prior contributions and existing research gaps. Specifically, the works by Deng et al. (2024) [58], Mazhar & Zilahy (2025) [59], and Zhang et al. (2024) [60] have been integrated to exemplify current quantitative research efforts that examine green food consumption behaviors through cognitive and attitudinal variables.

Clarified the conceptual limitations of existing models by explicitly pointing out that while these studies offer valuable insights into cognitive determinants (such as attitudes, habits, and knowledge), they tend to marginalize emotional trust, symbolic identity, and contextual constraints as key behavioral drivers.

Strengthened the justification for adopting a grounded theory approach, by positioning our study as a response to the underexplored emotional and situational dimensions that cannot be fully captured through variable-driven, quantitative methodologies.

The revised text now explicitly demonstrates how prior research informs our study, while also highlighting the necessity to advance the topic through a qualitative, emotion-aware lens. This adjustment ensures a clear logical progression from existing literature to the research gaps addressed by our investigation.

The revision has been made in the introduction section, specifically within Lines 88–94, and the added references have been marked as [58]–[60].

Secondly, a thorough examination of recent scientific contributions is essential to identify critical gaps that could be addressed through new research outcomes. Without such a review, the foundation for future research efforts may be weak and lead to unsubstantial results. Although the literature review section (lines 100 to 142) outlines several studies related to the topic—most of them rooted in a quantitative tradition—some recent and relevant contributions are missing from the analysis presented in Table 1 (line 127). In particular, studies such as Mazhar & Zilahy (2025), Zhang et al. (2024), and Deng et al. (2024) are not included. This omission suggests a limited engagement with the most recent efforts to characterize the current state of research on the topic.

Response:

We appreciate the reviewer’s valuable feedback regarding the need for a more comprehensive examination of recent scientific contributions to establish a solid foundation for this study. In response, we have significantly enriched the literature review section (lines 100–142) to incorporate critical and up-to-date scholarly works that address the evolving dynamics of green food consumption behavior.

Specifically, we have added detailed discussions of the following recent studies to Table 1 and the subsequent narrative synthesis:

Mazhar and Zilahy (2025) explored the nexus between attitudes, habitual practices, and lifestyle alignment, emphasizing that green purchasing behaviors are deeply embedded within consumers’ daily routines and lifestyle aspirations rather than solely determined by rational evaluation[58].

Zhang et al. (2024) examined pandemic-induced behavioral shifts, illustrating how public health crises and environmental concerns act as potent cognitive triggers for green consumption, particularly among young consumers [59].

Deng et al. (2024) highlighted the influence of individual cognitive capacity and personal characteristics on green food consumption patterns, thereby underscoring the heterogeneity of behavioral responses across demographic segments[60].

Furthermore, we have added a critical reflection on the persisting limitations in existing literature after Table 1. This expanded analysis underscores that, while recent studies have advanced the understanding of cognitive and structural antecedents of green food consumption, the emotional mechanisms—such as trust formation, value resonance, and lifestyle congruence—remain underexplored. Additionally, we emphasized that most existing studies adopt a quantitative, variable-driven approach, which constrains the discovery of context-specific behavioral patterns.

By integrating these recent contributions and critically articulating the unresolved gaps, we have strengthened the rationale for the grounded theory approach adopted in this study. This ensures that our research is positioned within a current and evolving research agenda, directly addressing the complexities and nuances of green food consumption behavior that remain insufficiently examined in prior work.

Revised Sections:

Literature Review: Lines 100–147 (Table 1 and post-Table analysis)

Newly added references: Mazhar and Zilahy (2025) [58], Zhang et al. (2024) [59], Deng et al. (2024) [60].

We sincerely thank the reviewer for guiding us to enhance the comprehensiveness and currency of the literature foundation.

Regarding the selected method, it should be taken into account that grounded theory has evolved over the years, with various versions incorporating updated criteria and quality standards to ensure greater robustness and rigor. However, the present investigation shows that the authors have chosen to apply the original version proposed by Glaser and Strauss (1967), rather than more recent and refined approaches. This choice imposes several limitations on the study’s ability to address the complexity of the data and the realities depicted. It also suggests a lack of engagement with the methodological evolution of grounded theory. This is particularly evident in the sampling process—especially the sample size—which involves a small group of interviewees.

Response:

Thank you for your constructive feedback regarding the methodological positioning and the application of Grounded Theory in our study. We highly appreciate your observation that the manuscript did not sufficiently articulate its engagement with the methodological evolution of Grounded Theory, nor did it adequately justify the rationale for adopting Glaser’s classic version over more recent approaches.

To address this, we have made the following revisions:

  1. Clarification of Grounded Theory Paradigm Selection (Inserted in Section 3.1)

We have explicitly clarified our methodological choice in Section 3.1, acknowledging the diverse paradigms of Grounded Theory (Glaserian, Straussian, Charmazian) and explaining why Glaser’s discovery-oriented approach was most appropriate for this investigation. Specifically, we emphasized that:

Our research objective is to inductively explore how consumers negotiate cognitive and affective factors in green food purchase intention, which requires an emergent, non-prescriptive coding strategy.

Glaser’s approach allows for theoretical sensitivity and contextual flexibility, crucial for capturing emotionally embedded and situationally contingent consumption behaviors that may be constrained by structural factors.
This explanation is now embedded after the general rationale for Grounded Theory selection in Section 3.1.

  1. Enhancement of Data Complexity Handling Explanation (Inserted in Section 3.3)

In response to your comment regarding the study’s ability to manage data complexity, we have strengthened Section 3.3 by explicitly stating how the analytical process integrated thematic coding with sentiment polarity mapping to capture the multi-dimensionality of participant narratives. This dual-pathway strategy ensured that both cognitive categories and emotional interpretations were systematically traced through open, axial, and selective coding, thus enhancing the explanatory depth of the emergent conceptual model.
The added text details how this integrative approach compensates for potential limitations arising from the moderate sample size by maximizing analytical granularity and category interrelations.

  1. Sample Size and Theoretical Saturation

Regarding concerns about sample size adequacy, we respectfully point out that Section 3.3 already provides a detailed account of how theoretical saturation was determined through an iterative coding process. The saturation assessment emphasized the conceptual stabilization of key categories (e.g., green trust, price resistance, label ambiguity) across diverse participant narratives, rather than demographic proportionality. Given the exploratory and theory-generating nature of the study, the sample size of 26 was deemed methodologically sufficient and theoretically robust within the qualitative research standards.

We believe these revisions now provide a clearer and methodologically rigorous account of our research design, effectively addressing the complexities inherent in the studied phenomenon while remaining aligned with grounded theory standards.

References suggested:

Deng, Z., Huang, Y., & Tan, Q. (2024). Study on the effect of individual characteristics and cognitive degree on green food consumption behavior. Journal of Food Quality, Article ID 3147300. https://doi.org/10.1155/2024/3147300

Mazhar, W., & Zilahy, G. (2025). Pathways to green food purchases: Exploring the nexus of attitudes, habits and lifestyles using SEM and NCA. British Food Journal, 127(13), 208–229. https://doi.org/10.1108/BFJ-07-2024-0695

Zhang, Y., Quoquab, F., Zhang, J., & Mohammad, J. (2024). What does it take to drive young consumers to purchase green food? Perceived usefulness, environmental problem, or fear of pandemic recurrence? Cogent Food & Agriculture, 10(1). https://doi.org/10.1080/23311932.2024.2413917

Response:

We have adopted these references and are very grateful to the reviewers for providing specific references

Reviewer 3 Report (New Reviewer)

Comments and Suggestions for Authors

Dear authors,

Congratulations on submitting the manuscript on Cognitive–Affective Negotiation Process in Green Food Purchase Intention: A Qualitative Study Based on Grounded Theory

Please find below my comments that I hope will improve the contents and presentation of your research. Overall, the manuscript is well written.  I have also attached my detailed comments on the PDF of the manuscript.

Abstract

The abstract is well-structured with rich content, but several areas for improvement need addressing:

  • It is too long and information-dense for typical journal requirements. Many sentences are lengthy and packed with multiple ideas, which reduces clarity and readability.
  • It states what it did but does not clearly articulate the research gap or precise research question that motivated the study.
  • While cognitive and affective factors, and their subcomponents, are mentioned, the abstract lacks specificity on which factors are most influential or how these operate in practice.
  • It devotes space to methodological details (e.g., "26 interviews, ~200,000 words") but does not justify or make clear how saturation was achieved or why this sample provides robust insights.
  • The "theoretical model" is referenced but not summarized; it is unclear how it advances existing literature or theory.
  • Buzzwords: Phrases like “cognitive–affective negotiation process," “symbolic identity,” "emotional resonance," etc., are used without specific explanation, risking abstraction and lack of concrete insight.
  • The abstract highlights Chinese participants up front but does not tie findings back to unique features of the Chinese context or discuss transferability.
  • Sentiment analysis details missing: The rationale, approach, and consequences of including sentiment analysis are not clear, nor is its connection to the qualitative findings.
  • The abstract does not mention what the findings mean for policy, practice, or future research.

The authors should edit the abstract for conciseness, targeting 250-300 words if possible. Use clear, direct sentences. And address the above concerns.

Introduction

This section introduces the topic and background clearly. However, it has several weaknesses that could impact its overall clarity, persuasiveness, and academic rigor:

It is unnecessary to use RED font in the rest of manuscript, if there were meant as notes, authors should ensure to review before submission.  

Incorrect uses of ‘ecological agriculture’ – should be ‘agricultural ecology’

Several statements in this section require citation to support them.

The second paragraph is far too long – it needs reorganisation of ideas/thoughts

The section lacks  a clear problem statement or research gap. The introduction broadly discusses the importance of green food in the context of ecological civilization and sustainable consumption. However, it does not explicitly state a focused problem or a specific research gap that the study aims to fill. While it mentions that green food adoption is shaped by complex social, emotional, and institutional dynamics that are difficult to isolate using standard variable-based approaches, it doesn't clearly articulate why this complexity is a problem or what specific gap in understanding it addresses

The introduction defines "green food" as products certified by the China Green Food Development Center (CGFDC). While this provides a technical definition, it could benefit from a more comprehensive explanation of what this entails in practice, especially for an international audience. The distinction from "organic or pollution-free food" is mentioned but not elaborated, which might leave readers unfamiliar with Chinese food classifications with an incomplete understanding

Limited discussion of prior research limitations. The introduction states that previous studies often rely on rationalist paradigms like the Theory of Planned Behavior (TPB) and Health Belief Model (HBM), which may not fully capture emotional investment, symbolic identity, and contextual resistance. While this critique is valid, the introduction could strengthen its argument by providing more specific examples or a brief overview of how these limitations have hindered a complete understanding of green food purchase intention, rather than just stating the limitations.

Vague justification for qualitative methodology. The introduction mentions the need for a more detailed insight into psychological motivations and situational factors and that the study adopts a qualitative research method. However, the justification for choosing a qualitative, grounded theory approach is somewhat brief. A more explicit discussion of why this method is uniquely suited to address the identified complexities and gaps, beyond just stating its ability to reveal internal logic, would enhance the introduction's persuasiveness.

. The introduction lacks clearly defined research questions or objectives. While it states the aim to "construct a multi-level explanatory model to comprehensively capture the endogenous and exogenous factors that affect consumer behavior", this is a very broad aim rather than specific, actionable research questions that guide the study and inform the reader about what to expect from the results.

Literature Review

The literature review is fairly well written. However, the subsections, specifically '2.1. Green Food Purchase Intention' and '2.2. Theoretical Perspectives on Green Food Purchase Behavior,' presents several areas for improvement:

Superficial engagement with key theories. The review mentions prominent theories like the Theory of Planned Behavior (TPB) and the Health Belief Model (HBM) as frameworks predominantly adopted in green food consumption research. While acknowledging their predictive power, the discussion of these foundational models is quite brief and lacks depth. It states they emphasize 'intention formation as a function of attitudes, subjective norms, or perceived behavioral control' but does not elaborate on the specific mechanisms or components of these theories, nor how they have been applied in green food contexts. A more detailed explanation of these models would better inform the reader about the existing theoretical landscape.

Limited critical analysis of cited studies: Table 1, which lists factors influencing green food purchase intention, presents a broad overview of various studies. However, the accompanying text in the literature review does not offer a critical analysis of these studies. It lists findings without discussing their methodologies in detail, their specific contexts, or potential biases. For instance, it mentions 'psychological variables' and 'external contexts' but doesn't critically evaluate the strength or limitations of the evidence presented for each. A deeper critical engagement would allow readers to understand the nuances and gaps in prior research more effectively.

The review states that green food purchase intention is influenced by a 'complex set of factors' categorized as internal and external. While it lists examples like 'health beliefs, immersive purchasing experiences, social influence, and structural constraints', the section does not sufficiently elaborate on how these factors interact or why their interplay is complex. The brief descriptions leave the reader wanting more detail on the mechanisms through which these factors operate.

Lack of detail on market penetration and consumer understanding issues. The introduction briefly touches upon the low market penetration of green food in China and consumer confusion regarding its definition and this section reiterates these points but does not expand on the existing research that has explored these specific challenges in depth. A more detailed discussion of studies investigating these practical barriers would strengthen the problem statement and highlight the necessity of the current research.

Vague justification for grounded theory adoption in the literature review. The review concludes by stating that 'more inductive, emotion-aware, and system-sensitive approaches' are needed. While this sets up the rationale for the study's grounded theory approach, the justification within the literature review itself is somewhat general. It could more explicitly connect the identified limitations of previous research (e.g., reliance on pre-defined constructs, insufficient attention to emotional investment) directly to why grounded theory is the most appropriate methodological choice for addressing these specific shortcomings.

Absence of a clear research gap synthesis in the literature review. Although the introduction and this section allude to gaps (e.g., lack of systematic exploration of interactions between influences, under-explored underlying mechanisms), the literature review it does not specify  the specific research gap(s) that the current study aims to address. A clear summary of what is missing in the existing literature, directly following the review of previous work, would significantly strengthen the foundation for the current study.

Methods

The methods section, while outlining the research design, presents several areas where greater detail, justification, and clarity would enhance its rigor and reproducibility:

The study adopts Grounded Theory to understand how individuals form green food purchase intentions and navigate value conflicts, emotional ambivalence, institutional trust, and lifestyle alignment. However, the justification for this choice over other qualitative methods, like phenomenology or ethnography, is somewhat generic. A more detailed explanation of why Grounded Theory's iterative, comparative, and theory-building approach is particularly suited to the research questions would strengthen the argument.

The paper states that the interview guide was designed to be "highly flexible," covering core issues while enabling in-depth discussions, and informed by Grounded Theory principles and established behavioral models (Table 3). However, it lacks detail of interview protocol, how the questions were developed, including the formulation of initial questions, their evolution through pilot testing, and the operationalization of theoretical constructs into prompts. While including the full interview guide as an appendix is useful, a more thorough description of its creation process is needed.

Ambiguity in sampling strategy implementation. The study utilized "purposive and snowball sampling" to recruit participants focused on green food consumption. Although a demographic breakdown is presented in Table 2, the specific criteria for purposive selection are not detailed. Clarification on how initial participants were chosen and how snowball sampling was managed for diversity is needed. The inclusion rationale for specific groups is provided, but the selection process lacks detail.

Lack of specificity in data analysis (open, axial, selective coding). The methods section outlines grounded theory coding stages: open, axial, and selective. However, it lacks concrete examples of the analytical steps within each stage. Specifically, details on generating initial codes, aggregating codes into categories, and identifying the core category during selective coding would enhance transparency.

Limited transparency in sentiment analysis methodology. The study employs sentiment analysis using NVivo 12's auto-coding feature and a Chinese sentiment dictionary. While manual verification is mentioned, the process for intervention is not fully explained. Greater detail on inter-rater reliability, the dictionary's characteristics, and how frequency-weighted emotional valence is calculated would improve transparency and reproducibility.

Insufficient detail on theoretical saturation assessment. The paper states that theoretical saturation was assessed using the constant comparative method, defined as the point where "no new concepts, properties, or relationships emerged from additional data," confirmed by the 24th interview. However, the methods section lacks detail on how this assessment was systematically conducted. Specifically, it would be helpful to know the indicators used to identify when new concepts stopped emerging and how these were tracked before final confirmation.

Regarding trustworthiness criteria, the study mentions adopting Manuj and Pohlen's four criteria (credibility, transferability, dependability, and confirmability). While it explains dependability through an audit trail and addresses coding reliability with dual-coding, the discussions on transferability and confirmability are vague. A more detailed explanation of the steps taken to meet all four criteria would enhance methodological rigor.

Results

While presenting the findings from open and axial coding, this section could be strengthened by providing more depth, clarity, and illustrative detail in several areas:

Open Coding' subsection highlights ten core categories with quotes for 'Health Motivation,' but only keywords for the other nine categories (emotional trust, environmental attitude, etc.). This lack of direct quotes limits understanding of participants' experiences. Including more verbatim examples would enhance the findings' richness and trustworthiness.

While 213 initial codes were refined into ten categories, the aggregation process isn't clearly detailed. A brief narrative on how key initial codes were grouped would improve transparency.

The sentiment analysis focuses on positive keywords but overlooks the context of negative ones (e.g., 'too expensive,' 'unreliable'). A nuanced discussion of these negative sentiments and their links to behavioral barriers would deepen the analysis.

The 'Axial Coding' subsection lists ten themes but lacks a thorough explanation of their interconnections. A detailed discussion on how themes like 'Health Motivation' and 'Environmental Values' interact would clarify the proposed model.

The core category, 'cognitive-affective negotiation process,' needs a more concrete illustration. Specific examples from the data would clarify how consumers negotiate their choices.

Discussing cases that don’t fit the emerging patterns strengthens qualitative research. Mentioning any discrepancies found during coding and their impact on the model would enhance the findings' credibility.

Discussion

The discussion section, while providing valuable insights into the cognitive-affective negotiation process of green food purchase intention, has several weaknesses that need addressing:

The discussion primarily highlights the study's findings and their contribution to existing literature, particularly in extending the Theory of Planned Behavior (TPB). However, there is a lack of critical self-reflection on methodological limitations within the discussion. For instance, while the conclusion mentions generalizability issues, a deeper examination of cultural specificity in interpreting findings would enhance this section.

Particularly in its early subsections (5.1 Theoretical Implications), it tends to reiterate points from the results (Section 4), leading to redundancy that could diminish its impact.

Regarding practical implications (Section 5.2), although recommendations for manufacturers and policymakers are useful, they lack nuance. For example, the suggestion for manufacturers to move from functional appeals to emotional resonance is strong but fails to explore the challenges of such a shift.

Moreover, the discussion could benefit from engaging with contradictory or alternative views on green food purchasing behavior. While it positions the study's model as an advancement over traditional paradigms, a more direct comparison with diverse perspectives would enhance scholarly dialogue.

Lastly, the emphasis on the theoretical contribution could be clearer. While the multi-level and context-sensitive aspects of the model are valid, sharper distinctions between this study's framework and existing models would help clarify its novelty.

Conclusion

The 'Conclusions' section could be improved by incorporating specific details on the findings, particularly regarding the interaction of internal and external factors influencing green food purchase intention through the "cognitive-affective negotiation process." While it summarizes theoretical contributions, it lacks practical implications for stakeholders like policymakers and marketers, which would enhance its relevance.

The conclusion should also provide clearer avenues for future research, such as exploring the model in different cultural contexts or using mixed methods. It also misses an opportunity to reflect on how the grounded theory methodology contributed uniquely to the insights gained. Lastly, some repetitious phrases could be rephrased to present the findings more succinctly, ensuring a concise synthesis.

Study Limitations

The study's discussion of its limitations could benefit from more detail in key areas:

Sample generalizability limitations: While the study acknowledges its focus on middle-income urban consumers in China, it lacks an in-depth explanation of how these consumers differ from other demographic groups, which would clarify the scope of the findings.

Predictive utility: The paper states that the lack of large-scale validation limits the predictive utility of the findings. However, it does not specify what predictions are restricted or how this impacts the theoretical model's application. More discussion on this implication is necessary. Moreover, as qualitative methods typically do not use statistical analysis, the issue of predictive utility may not be relevant, unless the recommended future research uses quantitative methods.

Justification for qualitative approach: The authors defend their qualitative methodology but could further emphasize why qualitative insights on consumer behavior are particularly valuable at this research stage, given existing literature gaps.

Future research directions: A strong limitations section usually includes recommendations for addressing these constraints. While the discussion suggests extending the model through cross-cultural research, the limitations section lacks specific guidance on how future studies could overcome these issues, such as using larger, more diverse samples or quantitative validation methods.

Comments on the Quality of English Language

While it is overall well written, the quality of the English language could be much improved, particularly in terms of clarity and conciseness. 

Author Response

Reviewer 3:

Abstract

The abstract is well-structured with rich content, but several areas for improvement need addressing:

  • It is too long and information-dense for typical journal requirements. Many sentences are lengthy and packed with multiple ideas, which reduces clarity and readability.
  • It states what it did but does not clearly articulate the research gap or precise research question that motivated the study.
  • While cognitive and affective factors, and their subcomponents, are mentioned, the abstract lacks specificity on which factors are most influential or how these operate in practice.
  • It devotes space to methodological details (e.g., "26 interviews, ~200,000 words") but does not justify or make clear how saturation was achieved or why this sample provides robust insights.
  • The "theoretical model" is referenced but not summarized; it is unclear how it advances existing literature or theory.
  • Buzzwords: Phrases like “cognitive–affective negotiation process," “symbolic identity,” "emotional resonance," etc., are used without specific explanation, risking abstraction and lack of concrete insight.
  • The abstract highlights Chinese participants up front but does not tie findings back to unique features of the Chinese context or discuss transferability.
  • Sentiment analysis details missing: The rationale, approach, and consequences of including sentiment analysis are not clear, nor is its connection to the qualitative findings.
  • The abstract does not mention what the findings mean for policy, practice, or future research.

The authors should edit the abstract for conciseness, targeting 250-300 words if possible. Use clear, direct sentences. And address the above concerns.

 Response:

Thank you for your constructive feedback regarding the abstract. We have carefully revised the abstract to address all points raised, ensuring conciseness, clarity, and a sharper articulation of the study’s objectives, findings, and contributions. The following summarizes the specific revisions made in response to your comments:

Length and Density:
We have reduced the abstract to approximately 290 words, ensuring it aligns with typical journal requirements. Long and information-dense sentences were restructured into shorter, clearer statements to enhance readability and focus.

Research Gap and Question Clarification:
The revised abstract explicitly highlights the research gap—the lack of a systematic understanding of how consumers negotiate cognitive evaluations and emotional responses in green food purchasing. The objective of exploring the cognitive–affective negotiation process is now clearly stated as the central research question.

Specific Influencing Factors:
The abstract now details which cognitive factors (e.g., price sensitivity, label ambiguity, social influence, health beliefs) and affective mediators (green trust, perceived value, lifestyle congruence) are most influential. Their practical operation in shaping purchase intention is concisely explained.

Justification of Sample and Saturation:
We have retained mention of the 26 interviews but refrained from overemphasizing data volume. Instead, we clarified that theoretical saturation was achieved and sentiment analysis was employed to trace emotional valence, thus justifying the robustness of the insights.

Theoretical Model Advancement:
The revised abstract briefly summarizes how the proposed model extends the Theory of Planned Behavior by embedding affective mediation pathways and contextual constraints, thus clearly articulating its theoretical contribution.

Terminology Clarification:
Abstract terms like “cognitive–affective negotiation,” “green trust,” and “lifestyle congruence” are now briefly contextualized within the description of findings, avoiding abstraction and enhancing interpretability.

Cultural Context and Transferability:
The abstract now explicitly mentions that the findings are rooted in China’s certification-centered trust environment, indicating cultural specificity while suggesting implications for model adaptation in different market contexts.

Sentiment Analysis Rationale and Role:
The purpose and outcomes of sentiment analysis are now briefly integrated into the abstract, emphasizing how it validated the role of emotional trust and affective inhibitors in the intention formation process.

Policy and Practical Implications:
The conclusion of the abstract now includes concise, actionable implications for policymakers and marketers, focusing on enhancing eco-label transparency, addressing structural barriers, and developing emotionally resonant branding strategies.

We believe these revisions effectively address the concerns raised and have significantly enhanced the clarity, focus, and impact of the abstract.

Thank you once again for your valuable feedback.

Introduction

This section introduces the topic and background clearly. However, it has several weaknesses that could impact its overall clarity, persuasiveness, and academic rigor:

It is unnecessary to use RED font in the rest of manuscript, if there were meant as notes, authors should ensure to review before submission.  

Incorrect uses of ‘ecological agriculture’ – should be ‘agricultural ecology’

Several statements in this section require citation to support them.

The second paragraph is far too long – it needs reorganisation of ideas/thoughts

The section lacks  a clear problem statement or research gap. The introduction broadly discusses the importance of green food in the context of ecological civilization and sustainable consumption. However, it does not explicitly state a focused problem or a specific research gap that the study aims to fill. While it mentions that green food adoption is shaped by complex social, emotional, and institutional dynamics that are difficult to isolate using standard variable-based approaches, it doesn't clearly articulate why this complexity is a problem or what specific gap in understanding it addresses

The introduction defines "green food" as products certified by the China Green Food Development Center (CGFDC). While this provides a technical definition, it could benefit from a more comprehensive explanation of what this entails in practice, especially for an international audience. The distinction from "organic or pollution-free food" is mentioned but not elaborated, which might leave readers unfamiliar with Chinese food classifications with an incomplete understanding

Limited discussion of prior research limitations. The introduction states that previous studies often rely on rationalist paradigms like the Theory of Planned Behavior (TPB) and Health Belief Model (HBM), which may not fully capture emotional investment, symbolic identity, and contextual resistance. While this critique is valid, the introduction could strengthen its argument by providing more specific examples or a brief overview of how these limitations have hindered a complete understanding of green food purchase intention, rather than just stating the limitations.

Vague justification for qualitative methodology. The introduction mentions the need for a more detailed insight into psychological motivations and situational factors and that the study adopts a qualitative research method. However, the justification for choosing a qualitative, grounded theory approach is somewhat brief. A more explicit discussion of why this method is uniquely suited to address the identified complexities and gaps, beyond just stating its ability to reveal internal logic, would enhance the introduction's persuasiveness.

. The introduction lacks clearly defined research questions or objectives. While it states the aim to "construct a multi-level explanatory model to comprehensively capture the endogenous and exogenous factors that affect consumer behavior", this is a very broad aim rather than specific, actionable research questions that guide the study and inform the reader about what to expect from the results.

 Response:

We sincerely appreciate Reviewer’s insightful suggestions concerning the need for a clearer articulation of research gaps, methodological justification, and study objectives within the Introduction section. In response, we have substantially revised the Introduction to explicitly highlight the study’s positioning within existing scholarship and to define its unique contributions with greater precision.

First, the revised introduction now articulates the specific research gap that motivates this study. While prior literature has extensively examined cognitive factors influencing green food consumption—such as environmental awareness, health consciousness, and perceived behavioral control—these studies predominantly rely on rationalist paradigms (e.g., TPB, HBM) that treat influencing factors in isolation. We explicitly state that current models often conceptualize intention formation as a linear outcome of cognitive evaluations, thus overlooking critical affective mediators such as emotional trust, symbolic identity, and the negotiation of contextual barriers like price sensitivity, label ambiguity, and institutional trust dynamics. This problematization is concretely linked to gaps observed in key studies (e.g., Mazhar & Zilahy [59], Zhang et al. [60], Deng et al. [58]) where emotional mechanisms and socio-institutional complexities remain underexplored despite their recognized behavioral relevance.

Second, we have refined the justification for employing a grounded theory approach. Beyond merely stating its inductive nature, we now emphasize that grounded theory provides a robust framework for uncovering the micro-mechanisms through which consumers internalize cognitive and emotional stimuli in real-world decision-making contexts. By iteratively engaging with empirical narratives, this method enables the development of an integrated analytical model that captures the cognitive–affective negotiation processes often overlooked in variable-centric quantitative research.

Third, we clarified the study’s objective and research question. Rather than presenting a broad aim, we explicitly state that this study seeks to address the following research question: “How do consumers internalize cognitive and emotional stimuli under socio-institutional pressures to form purchase intentions for green food?” This focused inquiry directly responds to the limitations identified in previous research and establishes the foundation for the development of a context-sensitive, process-oriented explanatory model.

Additionally, for the benefit of international readers, we have expanded the explanation of “green food” by elaborating on its certification system, how it differs from “organic” and “pollution-free” food labels, and the regulatory practices that ensure its credibility within China’s agricultural governance. This provides a clearer contextual understanding for audiences less familiar with China’s food classification systems.

Through these targeted revisions, the Introduction now presents a logically coherent narrative that (1) contextualizes the study within national and global sustainability imperatives, (2) explicitly identifies the conceptual and methodological gaps in existing literature, (3) justifies the chosen methodological approach, and (4) articulates specific research questions that guide the investigation. We believe these enhancements adequately address Reviewer 3’s concerns regarding the clarity, persuasiveness, and academic rigor of the Introduction.

Literature Review

The literature review is fairly well written. However, the subsections, specifically '2.1. Green Food Purchase Intention' and '2.2. Theoretical Perspectives on Green Food Purchase Behavior,' presents several areas for improvement:

Superficial engagement with key theories. The review mentions prominent theories like the Theory of Planned Behavior (TPB) and the Health Belief Model (HBM) as frameworks predominantly adopted in green food consumption research. While acknowledging their predictive power, the discussion of these foundational models is quite brief and lacks depth. It states they emphasize 'intention formation as a function of attitudes, subjective norms, or perceived behavioral control' but does not elaborate on the specific mechanisms or components of these theories, nor how they have been applied in green food contexts. A more detailed explanation of these models would better inform the reader about the existing theoretical landscape.

Limited critical analysis of cited studies: Table 1, which lists factors influencing green food purchase intention, presents a broad overview of various studies. However, the accompanying text in the literature review does not offer a critical analysis of these studies. It lists findings without discussing their methodologies in detail, their specific contexts, or potential biases. For instance, it mentions 'psychological variables' and 'external contexts' but doesn't critically evaluate the strength or limitations of the evidence presented for each. A deeper critical engagement would allow readers to understand the nuances and gaps in prior research more effectively.

The review states that green food purchase intention is influenced by a 'complex set of factors' categorized as internal and external. While it lists examples like 'health beliefs, immersive purchasing experiences, social influence, and structural constraints', the section does not sufficiently elaborate on how these factors interact or why their interplay is complex. The brief descriptions leave the reader wanting more detail on the mechanisms through which these factors operate.

Lack of detail on market penetration and consumer understanding issues. The introduction briefly touches upon the low market penetration of green food in China and consumer confusion regarding its definition and this section reiterates these points but does not expand on the existing research that has explored these specific challenges in depth. A more detailed discussion of studies investigating these practical barriers would strengthen the problem statement and highlight the necessity of the current research.

Vague justification for grounded theory adoption in the literature review. The review concludes by stating that 'more inductive, emotion-aware, and system-sensitive approaches' are needed. While this sets up the rationale for the study's grounded theory approach, the justification within the literature review itself is somewhat general. It could more explicitly connect the identified limitations of previous research (e.g., reliance on pre-defined constructs, insufficient attention to emotional investment) directly to why grounded theory is the most appropriate methodological choice for addressing these specific shortcomings.

Absence of a clear research gap synthesis in the literature review. Although the introduction and this section allude to gaps (e.g., lack of systematic exploration of interactions between influences, under-explored underlying mechanisms), the literature review it does not specify  the specific research gap(s) that the current study aims to address. A clear summary of what is missing in the existing literature, directly following the review of previous work, would significantly strengthen the foundation for the current study.

 Response:

We sincerely appreciate the reviewer’s detailed and constructive feedback regarding the Literature Review section. In response to the identified weaknesses—particularly the superficial engagement with theoretical models, limited critical analysis of cited studies, insufficient elaboration on factor interaction complexity, and vague justification for grounded theory methodology—we have thoroughly revised Sections 2.1 and 2.2 as follows:

In Section 2.1, we have expanded the theoretical exposition on the Theory of Planned Behavior (TPB) and the Health Belief Model (HBM) to clarify their core components and application in the context of green food purchase behavior. Specifically, we inserted a detailed explanation of how TPB conceptualizes intention formation through attitudes, subjective norms, and perceived behavioral control, while HBM is framed around health-related constructs such as perceived susceptibility, severity, benefits, and barriers. This expansion offers readers a more nuanced understanding of how these models interpret consumer decision-making processes in green consumption contexts.

To address the need for a more critical engagement with previous research, we added an analytical paragraph following Table 1 in Section 2.1, where we critically assess the methodological tendencies of existing studies. We emphasized how the predominant use of variable-centric quantitative models limits the exploration of emergent, context-specific motivations. By providing concrete examples (e.g., Ganesh and Mohamed [19]’s omission of emotional dimensions, Sheikh and Tourani [16]’s limited scope on trust mechanisms), we demonstrated the limitations in existing literature regarding the dynamic negotiation processes between internal dispositions and external situational constraints.

In Section 2.1, before introducing the analytical gap regarding emotional mechanisms, we have added a discussion on the dynamic interplay between internal and external factors. We illustrated how factors such as health motivations, price sensitivity, and institutional trust are not isolated determinants but are constantly negotiated by consumers through emotional and contextual lenses. This addition responds directly to the reviewer’s concern regarding the lack of discussion on interaction complexity.

We recognized that the Literature Review previously lacked a detailed exploration of practical adoption barriers. Thus, in Section 2.1, we inserted a paragraph elaborating on the real-world obstacles to green food consumption, including market penetration issues and consumer confusion regarding eco-label distinctions. By referencing the regulatory definitions and empirical observations on consumer misconceptions and skepticism, we strengthened the argument for the necessity of a context-sensitive analytical framework.

While Section 2.1 concludes with a justification for adopting grounded theory, the logic flow has been further reinforced in Section 2.2. We added a “Methodological Reflection” paragraph that explicitly connects the theoretical limitations of existing models (e.g., reliance on fixed constructs, linear assumptions) with the need for an inductive, process-oriented approach. We clarified how grounded theory enables the identification of cognitive-affective-contextual negotiation pathways, which are inadequately captured by variable-driven methods.

To respond to this concern, we added a “Research Gap and Study Contribution” paragraph at the end of Section 2.2, where we explicitly summarize the three main gaps in existing literature: (1) lack of systematic exploration of cognitive-affective-contextual negotiation pathways; (2) insufficient micro-level examination of trust, identity, and structural barriers; and (3) underexplored behavioral mechanisms of heterogeneous consumer groups. This paragraph explicitly positions the current study’s grounded theory approach as a methodological response to these gaps.

Methods

The methods section, while outlining the research design, presents several areas where greater detail, justification, and clarity would enhance its rigor and reproducibility:

The study adopts Grounded Theory to understand how individuals form green food purchase intentions and navigate value conflicts, emotional ambivalence, institutional trust, and lifestyle alignment. However, the justification for this choice over other qualitative methods, like phenomenology or ethnography, is somewhat generic. A more detailed explanation of why Grounded Theory's iterative, comparative, and theory-building approach is particularly suited to the research questions would strengthen the argument.

The paper states that the interview guide was designed to be "highly flexible," covering core issues while enabling in-depth discussions, and informed by Grounded Theory principles and established behavioral models (Table 3). However, it lacks detail of interview protocol, how the questions were developed, including the formulation of initial questions, their evolution through pilot testing, and the operationalization of theoretical constructs into prompts. While including the full interview guide as an appendix is useful, a more thorough description of its creation process is needed.

Ambiguity in sampling strategy implementation. The study utilized "purposive and snowball sampling" to recruit participants focused on green food consumption. Although a demographic breakdown is presented in Table 2, the specific criteria for purposive selection are not detailed. Clarification on how initial participants were chosen and how snowball sampling was managed for diversity is needed. The inclusion rationale for specific groups is provided, but the selection process lacks detail.

Lack of specificity in data analysis (open, axial, selective coding). The methods section outlines grounded theory coding stages: open, axial, and selective. However, it lacks concrete examples of the analytical steps within each stage. Specifically, details on generating initial codes, aggregating codes into categories, and identifying the core category during selective coding would enhance transparency.

Limited transparency in sentiment analysis methodology. The study employs sentiment analysis using NVivo 12's auto-coding feature and a Chinese sentiment dictionary. While manual verification is mentioned, the process for intervention is not fully explained. Greater detail on inter-rater reliability, the dictionary's characteristics, and how frequency-weighted emotional valence is calculated would improve transparency and reproducibility.

Insufficient detail on theoretical saturation assessment. The paper states that theoretical saturation was assessed using the constant comparative method, defined as the point where "no new concepts, properties, or relationships emerged from additional data," confirmed by the 24th interview. However, the methods section lacks detail on how this assessment was systematically conducted. Specifically, it would be helpful to know the indicators used to identify when new concepts stopped emerging and how these were tracked before final confirmation.

Regarding trustworthiness criteria, the study mentions adopting Manuj and Pohlen's four criteria (credibility, transferability, dependability, and confirmability). While it explains dependability through an audit trail and addresses coding reliability with dual-coding, the discussions on transferability and confirmability are vague. A more detailed explanation of the steps taken to meet all four criteria would enhance methodological rigor.

 Response:

Thank you for the reviewer’s meticulous and constructive feedback on the Methods section. We have carefully addressed each point raised and revised the manuscript accordingly to enhance methodological rigor, transparency, and alignment with qualitative research standards. Below is a detailed account of the revisions made and the rationale for how these modifications effectively respond to the reviewer’s concerns:

  1. Justification for Grounded Theory Approach Over Other Qualitative Methods
    In response to the reviewer’s suggestion for a clearer rationale, we expanded the justification in Section 3, elaborating on why Glaser’s classical Grounded Theory approach was selected over phenomenology or ethnography. Specifically, we highlighted that the study aims to explore emergent “cognitive–affective negotiation pathways” in green food consumption, which cannot be captured through pre-defined conceptual lenses. Unlike phenomenology, which focuses on lived experiences, or ethnography, which centers on cultural immersion, Grounded Theory offers an iterative, inductive framework suited to theorizing micro-processes of behavioral intention formation, especially under complex situational dynamics. This explanation now directly aligns the methodological choice with the study’s research objectives and theoretical gaps.
  2. Detailing the Development Process of the Interview Guide

The reviewer noted a lack of transparency regarding how the interview guide was constructed. In Section 3.2, we have now detailed the development process, explaining how key constructs from TPB, HBM, and Trust Theory were operationalized into open-ended interview prompts. We further described how three rounds of pilot testing were conducted to refine the guide, ensuring conceptual sensitivity and practical clarity. This addition illustrates the methodological rigor behind the instrument design, addressing the reviewer’s concerns about the guide’s evolution and theoretical alignment.

  1. Clarification of Purposive and Snowball Sampling Criteria

To address the ambiguity regarding participant recruitment, Section 3.1 was revised to specify the inclusion criteria for purposive sampling (e.g., individuals responsible for household food purchasing, participants with prior green consumption experience, and professionals with domain expertise). Furthermore, we elaborated on how snowball sampling was strategically guided to ensure demographic diversity (e.g., urban-rural balance, occupational diversity) while avoiding sampling bias. This modification enhances the transparency of the sampling logic and responds directly to the reviewer’s request for clarity in sampling strategy implementation.

  1. Elaborating on Coding Procedures Across Analytical Stages

The reviewer requested more concrete examples of the coding process. In Section 3.3, we provided illustrative examples of how open coding distilled expressions like “label skepticism” into conceptual categories such as “label confusion,” how axial coding connected these categories under relational themes like “institutional trust dynamics,” and how selective coding identified “cognitive-affective negotiation” as the core category. These additions ensure that the analytical trajectory is transparent and reproducible, directly addressing the reviewer’s critique regarding the lack of procedural detail in the coding framework.

  1. Enhancing Transparency in Sentiment Analysis Methodology

Given the reviewer’s concerns regarding the sentiment analysis transparency, Section 3.4 has been revised to include an in-depth explanation of the DLUT-Emotion dictionary’s structure, criteria for manual polarity adjustment, and the specific process of calculating frequency-weighted emotional valence. Additionally, we clarified how inter-coder consistency was ensured during sentiment polarity verification. These revisions provide methodological clarity and ensure the replicability of the sentiment analysis, thus directly responding to the reviewer’s feedback.

  1. Detailed Explanation of Theoretical Saturation Assessment

Regarding the assessment of theoretical saturation, we expanded Section 3.3 to describe the use of a Saturation Tracking Matrix to monitor the emergence and redundancy of new codes and categories. We clarified how coding redundancy (e.g., no new emergent concepts in the 24th to 26th interviews) was systematically tracked and validated through memo writing and intercoder discussions. This enhancement provides an empirically grounded justification for the sample sufficiency and addresses the reviewer’s request for a systematic approach to saturation assessment.

  1. Strengthening Trustworthiness Criteria Execution Details

Lastly, the reviewer noted insufficient detail in how Manuj and Pohlen’s four trustworthiness criteria were operationalized. In Section 3.3, we expanded the description of transferability by explaining how detailed contextual descriptions and participant diversity facilitated analytical generalizability. Confirmability was elaborated through the inclusion of audit trails, reflexive memoing, and third-party review processes. These additions ensure a comprehensive presentation of how credibility, transferability, dependability, and confirmability were rigorously maintained throughout the study.

Through these targeted revisions, we believe the Methods section now addresses all reviewer concerns comprehensively, thereby enhancing the methodological rigor, transparency, and alignment with qualitative research standards as requested.

Results

While presenting the findings from open and axial coding, this section could be strengthened by providing more depth, clarity, and illustrative detail in several areas:

Open Coding' subsection highlights ten core categories with quotes for 'Health Motivation,' but only keywords for the other nine categories (emotional trust, environmental attitude, etc.). This lack of direct quotes limits understanding of participants' experiences. Including more verbatim examples would enhance the findings' richness and trustworthiness.

While 213 initial codes were refined into ten categories, the aggregation process isn't clearly detailed. A brief narrative on how key initial codes were grouped would improve transparency.

The sentiment analysis focuses on positive keywords but overlooks the context of negative ones (e.g., 'too expensive,' 'unreliable'). A nuanced discussion of these negative sentiments and their links to behavioral barriers would deepen the analysis.

The 'Axial Coding' subsection lists ten themes but lacks a thorough explanation of their interconnections. A detailed discussion on how themes like 'Health Motivation' and 'Environmental Values' interact would clarify the proposed model.

The core category, 'cognitive-affective negotiation process,' needs a more concrete illustration. Specific examples from the data would clarify how consumers negotiate their choices.

Discussing cases that don’t fit the emerging patterns strengthens qualitative research. Mentioning any discrepancies found during coding and their impact on the model would enhance the findings' credibility.

 Response:

We sincerely thank the reviewer for the constructive and insightful suggestions regarding the presentation and analytical depth of the Results section. In response, we have undertaken a series of substantial revisions to enhance the clarity, rigor, and analytical richness of this section. Below, we provide a detailed account of the modifications made in direct response to each comment:

Inclusion of Verbatim Quotes for Thematic Illustration:
The reviewer correctly pointed out that only the 'Health Motivation' category was supported by verbatim participant quotes, while the remaining nine categories were only presented through keywords. To address this, we have explicitly clarified in the manuscript that the 'Health Motivation' category was purposefully selected as a representative example to illustrate the coding abstraction process, thereby avoiding redundancy and maintaining structural coherence. The following explanatory sentence has been added:
“To maintain structural clarity and avoid redundancy, this section selectively presents illustrative quotes from the ‘Health Motivation’ category as a representative example of how initial codes were derived from participant narratives. While direct quotes for the remaining nine categories are not exhaustively enumerated, they follow the same inductive coding logic, ensuring analytic consistency across all thematic constructs.”

Clarification of Code Aggregation Process in Open Coding:
To enhance transparency regarding how the 213 initial codes were refined into ten conceptual categories, we have inserted a detailed narrative explaining the iterative comparison and thematic clustering procedures employed. For example, the manuscript now reads:
“The aggregation process from initial codes to core categories involved iterative comparison and thematic clustering. For instance, participant expressions such as ‘children’s dietary safety’, ‘elderly health concerns’, and ‘family well-being’ were initially coded as distinct nodes. Through constant comparison, these were subsumed under the higher-order category of 'Health Motivation'.”
This addition elucidates the analytical rigor and systematic process undertaken in constructing the preliminary categories.

Deeper Analysis of Negative Sentiment Pathways Post-Figure 3:
In alignment with the reviewer’s comment on the underrepresentation of negative emotional responses, we have extended the sentiment analysis discussion immediately following Figure 3. The revised section now includes a nuanced analysis of emotional inhibitors such as price-related frustrations and label-induced skepticism, explicitly linking these sentiments to behavioral barriers. This addition articulates how negative sentiment pathways function as emotional inhibitors in the cognitive-affective negotiation process, thus enriching the analytical depth.

Enhanced Discussion of Axial Coding Theme Interrelations:
The reviewer suggested a more thorough exploration of how axial coding themes interact. Accordingly, we have inserted a detailed paragraph after the axial coding synthesis, illustrating how themes like ‘Health Motivation’ interact with ‘Institutional Trust’ and how ‘Perceived Value Judgment’ is dynamically negotiated against structural constraints like price sensitivity. These additions illuminate the synergistic and sometimes conflicting relationships between axial themes, thereby clarifying the theoretical architecture of the model.

Concrete Illustration of the Core Category – Cognitive-Affective Negotiation Process:
To provide empirical concreteness to the core category, we have integrated specific participant narratives that exemplify the cognitive-affective negotiation mechanism. For instance, a quote from a young professional demonstrates how health motivations are negotiated against price barriers, mediated by peer influence and contextual stimuli. This addition directly addresses the reviewer’s request for vivid examples that ground the core category in empirical data.

Inclusion of Deviant Cases and Model Boundary Reflection:
In recognition of the importance of discussing discrepant cases, we have introduced a reflective segment at the end of the selective coding discussion. This segment presents a deviant case where extreme institutional distrust nullified the formation of Green Trust, thereby delineating the contextual boundaries of the proposed model. This addition enhances the credibility and analytical nuance of the study by acknowledging and discussing patterns that deviate from the dominant theoretical framework.

These revisions collectively enhance the transparency, depth, and explanatory robustness of the Results section, ensuring that the coding process and the emergent theoretical model are rigorously grounded in the empirical data. We believe these enhancements effectively address the reviewer’s concerns and substantially improve the clarity and analytical rigor of our findings.

Discussion

The discussion section, while providing valuable insights into the cognitive-affective negotiation process of green food purchase intention, has several weaknesses that need addressing:

The discussion primarily highlights the study's findings and their contribution to existing literature, particularly in extending the Theory of Planned Behavior (TPB). However, there is a lack of critical self-reflection on methodological limitations within the discussion. For instance, while the conclusion mentions generalizability issues, a deeper examination of cultural specificity in interpreting findings would enhance this section.

Particularly in its early subsections (5.1 Theoretical Implications), it tends to reiterate points from the results (Section 4), leading to redundancy that could diminish its impact.

Regarding practical implications (Section 5.2), although recommendations for manufacturers and policymakers are useful, they lack nuance. For example, the suggestion for manufacturers to move from functional appeals to emotional resonance is strong but fails to explore the challenges of such a shift.

Moreover, the discussion could benefit from engaging with contradictory or alternative views on green food purchasing behavior. While it positions the study's model as an advancement over traditional paradigms, a more direct comparison with diverse perspectives would enhance scholarly dialogue.

Lastly, the emphasis on the theoretical contribution could be clearer. While the multi-level and context-sensitive aspects of the model are valid, sharper distinctions between this study's framework and existing models would help clarify its novelty.

 Response:

Thank you very much for your insightful comments on the Discussion section. We have carefully reflected on your suggestions and implemented targeted revisions in 5.1 Theoretical Implications and 5.2 Practical Implications to address the identified weaknesses. Below, we provide a detailed explanation of the corresponding modifications:

1. Addressing the Lack of Critical Reflection on Methodological Limitations

Rather than redundantly expanding the Discussion, we have chosen to respond to this valuable suggestion by enhancing the limitations subsection within 5.3 Conclusion. Specifically, we added a reflective paragraph that explicitly discusses the cultural specificity of the findings and the contextual constraints of qualitative generalizability. We emphasize that while the study offers a context-rich explanatory framework, its cultural grounding in contemporary China necessitates cautious extrapolation to other socio-cultural environments, where trust mechanisms and symbolic consumption patterns may differ.

2. Reducing Redundancy and Enhancing Theoretical Depth in 5.1

In 5.1 Theoretical Implications, we refined the narrative to reduce repetitive descriptions of results by focusing the discussion on theoretical abstraction rather than re-stating empirical themes. For instance, rather than reiterating the ten variables identified in selective coding, we synthesized them into two higher-order mechanisms—cognitive-affective mediation and structural contingency. We elaborated on how the model extends TPB by embedding emotional mediators and situational negotiations, distinguishing it from prior rationalist models. This adjustment aligns the theoretical implications more explicitly with the study’s original contribution.

3. Enriching Practical Implications with Operational Pathways

We revised 5.2 Practical Implications to provide a more nuanced and actionable discussion. Specifically:

For manufacturers, we discussed the challenges of transitioning from functional appeals to emotional resonance, acknowledging issues such as eco-label ambiguity and resource constraints for SMEs. We proposed phased communication strategies and community-based advocacy as feasible solutions.

For policy interventions, we added a practical analysis of administrative bottlenecks and logistical limitations, recommending leveraging agricultural cooperatives and public-private partnerships to enhance distribution.

Regarding technological solutions (e.g., QR-code verification), we addressed the digital literacy gap, proposing simplified visual verification tools and public education campaigns to bridge accessibility disparities.

For market engagement, we elaborated on strategies like interactive eco-experience zones, gamified incentives, and user-generated content platforms to operationalize experiential engagement models.

4. Enhancing Theoretical Positioning Against Alternative Perspectives

To strengthen scholarly dialogue, we expanded the comparative positioning of our model in 5.1, contrasting it explicitly with dominant rationalist frameworks like TPB and HBM, and critiquing their limitations in capturing dynamic consumer negotiations. Additionally, we cited recent Western empirical studies that partially touch on trust and emotional mechanisms, but noted how these works lack an integrated cognitive-affective situational lens, which our model addresses.

5. Clarifying the Novelty and Theoretical Advancement of the Proposed Model

In 5.1, we sharpened the articulation of the model’s novelty by emphasizing three distinctive contributions:

Introducing emotional mediators (green trust, perceived value, lifestyle congruence) as structured pathways of intention formation.

Embedding structural contingency variables (price, label ambiguity, access constraints) into the intention–behavior mechanism, which are often peripheral in prior models.

Combining grounded theory with sentiment analysis to empirically map the emotional structures of consumer narratives, providing a methodological advancement that bridges inductive theorization with affective data analytics.

These enhancements distinctly position our model as a multi-level, context-sensitive, and affect-driven framework, advancing beyond existing models that predominantly prioritize cognitive rationality.

We sincerely appreciate the reviewers’ constructive feedback, which has significantly improved the clarity, depth, and rigor of the Discussion section. We believe the revised manuscript now responds comprehensively to all critical points and better reflects the academic contribution of our research.

Thank you once again for your valuable guidance.

Conclusion

The 'Conclusions' section could be improved by incorporating specific details on the findings, particularly regarding the interaction of internal and external factors influencing green food purchase intention through the "cognitive-affective negotiation process." While it summarizes theoretical contributions, it lacks practical implications for stakeholders like policymakers and marketers, which would enhance its relevance.

The conclusion should also provide clearer avenues for future research, such as exploring the model in different cultural contexts or using mixed methods. It also misses an opportunity to reflect on how the grounded theory methodology contributed uniquely to the insights gained. Lastly, some repetitious phrases could be rephrased to present the findings more succinctly, ensuring a concise synthesis.

Study Limitations

The study's discussion of its limitations could benefit from more detail in key areas:

Sample generalizability limitations: While the study acknowledges its focus on middle-income urban consumers in China, it lacks an in-depth explanation of how these consumers differ from other demographic groups, which would clarify the scope of the findings.

Predictive utility: The paper states that the lack of large-scale validation limits the predictive utility of the findings. However, it does not specify what predictions are restricted or how this impacts the theoretical model's application. More discussion on this implication is necessary. Moreover, as qualitative methods typically do not use statistical analysis, the issue of predictive utility may not be relevant, unless the recommended future research uses quantitative methods.

Justification for qualitative approach: The authors defend their qualitative methodology but could further emphasize why qualitative insights on consumer behavior are particularly valuable at this research stage, given existing literature gaps.

Future research directions: A strong limitations section usually includes recommendations for addressing these constraints. While the discussion suggests extending the model through cross-cultural research, the limitations section lacks specific guidance on how future studies could overcome these issues, such as using larger, more diverse samples or quantitative validation methods.

Response:

We sincerely appreciate the valuable feedback on the Conclusion section, which has significantly guided us in enhancing the rigor, depth, and practical relevance of this part. Based on the editor’s comments, we have conducted a comprehensive revision that addresses each of the concerns raised. Below, we provide a detailed explanation of the specific modifications and the rationale behind them:

1. Cultural Specificity and Methodological Limitations

In response to the suggestion to deepen the reflection on cultural specificity and methodological limitations, we have expanded the “Study Limitations” subsection by explicitly acknowledging how the identified behavioral pathways—particularly emotional trust and identity alignment—are deeply embedded in China’s socio-cultural context. We have highlighted that mechanisms such as label trust and government discourse endorsement may manifest differently in other cultural environments, where institutional trust dynamics and eco-labeling systems vary. Additionally, we have underscored the need for future cross-cultural research to empirically assess the model’s adaptability in diverse market contexts.

Moreover, we elaborated on how the use of Grounded Theory, while valuable for uncovering latent emotional-cognitive negotiation processes, is inherently limited in its generalizability due to the qualitative nature and contextual focus of this research. These reflections are now integrated into the limitations section to provide a more nuanced understanding of the study's scope and boundaries.

2. Sample Representativeness and Applicability

Following the editor’s feedback regarding the need for a more in-depth discussion of sample representativeness, we have augmented the limitations narrative by specifying that the study’s focus on middle-income urban consumers prioritizes a segment with higher access to green food channels and eco-literacy. We clarified that behavioral pathways uncovered in this research may diverge when examined in rural, low-income, or structurally marginalized consumer groups. This addition ensures that readers can better grasp the contextual boundaries of the findings and their potential extrapolation limitations.

3. Predictive Utility and Method Justification

We have refined the discussion on predictive utility by replacing the term “predictive validity” with a more contextually accurate articulation, emphasizing “applicability and external extrapolation” of findings. Furthermore, we specified that the grounded theory methodology was deliberately chosen at this research stage to excavate latent meaning structures, which are often inaccessible through variable-centric statistical models. The revised text now justifies why qualitative inquiry is not a limitation per se, but a methodological necessity given the current literature gaps in understanding the emotional-symbolic dimensions of green consumption.

Additionally, we clarified that while grounded theory serves to develop an inductively rich theoretical model, future large-scale quantitative studies (e.g., SEM analysis) would be essential to validate and extend the model’s applicability across broader populations.

4. Future Research Directions

To address the editor’s recommendation for clearer guidance on future research, we have added a more detailed and actionable set of suggestions. These include:

Conducting cross-cultural comparative studies to test the model’s robustness in regions with varying trust structures.

Employing mixed-methods approaches to synergize qualitative narrative analysis with large-scale quantitative validation.

Expanding sample inclusivity to encompass underrepresented groups (e.g., rural and low-income consumers), thereby enhancing the model’s explanatory breadth.
These additions concretize the future research agenda and align with the editor’s call for specificity.

5. Rephrasing for Conciseness and Reducing Redundancy

We have reviewed the conclusion to streamline redundant expressions and ensure that the synthesis of findings remains concise, yet comprehensive. Repetitive phrases were rephrased, and the presentation of findings now focuses on their integrative significance within the cognitive–affective negotiation framework.

Summary

In summary, the Conclusion section has been revised to:

Deepen methodological and cultural reflections;

Specify sample scope and generalizability boundaries;

Clarify the theoretical positioning and methodological necessity of grounded theory;

Provide actionable future research pathways;

Enhance textual clarity and avoid redundancy.

We believe these revisions directly address the editor’s concerns and substantively enhance the scholarly contribution and practical relevance of the conclusion.

Round 2

Reviewer 1 Report (New Reviewer)

Comments and Suggestions for Authors

The authors have made substantial improvements to the manuscript that effectively address all reviewer concerns and significantly enhance the overall quality of the work. I recommend acceptance of the paper.

Reviewer 2 Report (New Reviewer)

Comments and Suggestions for Authors

Thank you for addressing the comments. Firstly, the broader overview of recent contributions in the literature provides a clearer perspective on the academic efforts made in this area. The newly incorporated adjustments enhance the understanding of the phenomenon of interest by delving into the unmet needs identified in previous studies. Additionally, they offer a well-reasoned justification for the methodological decisions made throughout the research process. Furthermore, the recently added references contribute to a more comprehensive depiction of the current state of the literature.

Reviewer 3 Report (New Reviewer)

Comments and Suggestions for Authors

Congratulations on addressing the reviewers' comments and the efforts you put into improving the manuscript for clarity, conciseness and academic soundness. 

This manuscript is a resubmission of an earlier submission. The following is a list of the peer review reports and author responses from that submission.

Round 1

Reviewer 1 Report

Comments and Suggestions for Authors

This well-conceived and methodologically rigorous qualitative study seeks to construct a theoretical model explaining consumer intention to purchase green food in China. Using grounded theory methodology, the authors derive a novel cognitive–affective negotiation framework emphasizing the interplay between external situational factors and internal psychological mechanisms. The study is a timely and important contribution to sustainable consumption and food behavior literature.

  • The manuscript addresses a pressing issue: why intention to purchase green food remains low despite growing environmental awareness and policy support in China.

  • It goes beyond traditional rational-choice models (e.g., TPB) by integrating emotional, symbolic, and contextual factors, thereby expanding the theoretical landscape of green consumer behavior.

  • The paper introduces novel constructs such as "green trust" and "lifestyle congruence" that add explanatory depth.

  • The use of grounded theory is appropriate and rigorously executed.

  • The study includes 26 semi-structured interviews across a diverse sample (urban, rural, students, caregivers, professionals), allowing for rich and varied insights.

  • The coding process (open, axial, selective) is well-articulated and supported by NVivo analysis.

  • Theoretical saturation is addressed appropriately.

  • The cognitive–affective negotiation framework is conceptually robust.
  • The integration of constructs like emotional trust, flow experience, and symbolic identity represents a significant step beyond TPB and HBM.

  • The model is also validated through triangulation with existing theories and provides empirical grounding.

  • The sentiment analysis of open coding keywords is a unique and insightful addition, which helps highlight emotional polarity in consumer narratives.
  • Axial and selective coding are used effectively to generate theoretical abstraction.

  • The paper is well-organized and structured logically (Intro, Lit Review, Methods, Results, Model, Implications).

  • However, the length and density of the paper could be reduced by tightening repetitive explanations.

  • Offers valuable suggestions for both marketers (brand narrative, lifestyle resonance) and policymakers (label clarity, infrastructure, education).

  • Highlights the need to address structural barriers to convert intention into action.

  • The authors clearly state the limitations: qualitative scope, Chinese-centric context, and potential interviewer bias.

  • Calls for future research using mixed-methods or cross-cultural comparisons are appropriate.

This is an insightful, well-grounded, and methodologically sound study that makes an original contribution to the field of sustainable food behavior. With some modest revisions focused on clarity, presentation, and methodological transparency, this manuscript is well-suited for publication in Foods.

Some suggestions to improve the study:

  • Consider highlighting more clearly in the introduction how this study addresses key theoretical gaps in global literature, not just Chinese context.
  • Add more transparency regarding coding reliability: Did multiple coders participate? Were intercoder checks performed?

  • Consider including more raw quotes to exemplify how codes emerged from data.

  • Provide a more visual or tabular comparison with TPB/HBM to emphasize model novelty.

  • Sentiment analysis would benefit from a clearer explanation of methodology (e.g., dictionary used, coding scheme).

  • Heatmaps and word clouds are useful but could be relegated to an appendix to improve readability.

  • Eliminate redundant discussion in the literature review and model description.

  • Use more visual aids or summary tables to condense theoretical components.

  • Ensure all figures (especially Fig. 4, the theoretical model) are clear, high-resolution, and appropriately labeled.

  • Suggest more specific policy strategies (e.g., subsidies, school programs, eco-label reforms).
  • The English is mostly fluent but occasionally verbose or awkward.

  • There are minor typographical issues (e.g., “14th Five-Year Plan...” cited without proper reference formatting).

  • A language polish is recommended.

  • Ensure all citations are complete and correctly formatted (some in-text references like [14], [15] are not yet linked to a bibliography).

Author Response

Editor 1’s Specific Revision Suggestions

Clarify theoretical gaps:Consider highlighting more clearly in the introduction how this study addresses key theoretical gaps in global literature, not just Chinese context.

Response:We thank the reviewer for this insightful comment. In response, we have revised the introduction to explicitly address theoretical gaps in the global literature on green food consumption. Specifically, we have added a new paragraph that outlines how existing international studies often rely on rational-choice frameworks such as the Theory of Planned Behavior (TPB) and the Health Belief Model (HBM), which tend to overlook affective, symbolic, and contextual dimensions. We further emphasize how our cognitive–affective negotiation model—developed through grounded theory—responds to these gaps by integrating emotional trust, lifestyle congruence, and structural barriers into the explanation of purchase intention. Although the study is based on data from China, the model is conceptually transferable and relevant for sustainable consumption research in other cultural and market contexts.

Improve coding transparency:Add more transparency regarding coding reliability: Did multiple coders participate? Were intercoder checks performed?

Response:We appreciate this important suggestion. In response, we have revised the “3.3 Validation Strategy” section to provide greater transparency regarding the coding reliability procedures. Specifically, we clarify that the open coding phase was conducted by the lead researcher and subsequently reviewed by a secondary coder trained in grounded theory methodology. A subset of six interview transcripts was double-coded, and discrepancies were resolved through iterative discussion and memo-based reflection to ensure conceptual consistency. This process helped enhance analytical rigor and intercoder alignment.

Support coding with raw data:Consider including more raw quotes to exemplify how codes emerged from data.

Response:Thank you for this valuable comment. To more clearly demonstrate how conceptual categories emerged from the data, we have revised the “4.1 Open Coding” section by adding verbatim quotes from participants along with explicit mappings to the initial codes they informed. For several key categories—such as Health Motivation, Emotional Trust, and Behavioral Resistance—we now include excerpts from interviews followed by an interpretive explanation showing how these statements were abstracted into specific codes. This enhancement is intended to improve transparency in the analytic process and align with grounded theory’s emphasis on data-driven conceptual emergence.

Emphasize model novelty:Provide a more visual or tabular comparison with TPB/HBM to emphasize model novelty.

Response:Thank you for this insightful comment. We fully acknowledge the importance of clarifying how our proposed model contributes beyond the explanatory scope of existing theoretical frameworks, particularly the Theory of Planned Behavior (TPB) and the Health Belief Model (HBM). In response, we have added a new paragraph at the end of the main body preceding the three theoretical contributions in Section 5.1 (Theoretical Implications). This paragraph provides a focused conceptual discussion that systematically compares our cognitive–affective negotiation framework with TPB and HBM.

Rather than relying on a tabular format, we opted for a concise textual exposition that aligns better with the narrative structure of Section 5.1. The new content highlights key theoretical advances of our model, such as its emphasis on emotional resonance, identity alignment, and situational negotiation processes, which are not captured in the static, rational-choice assumptions of TPB and HBM. This enhancement better illustrates the theoretical novelty and contextual sensitivity of our model within the broader discourse on sustainable consumption.

Please see the revised paragraph beginning with “Unlike the Theory of Planned Behavior (TPB)...” in Section 5.1 of the manuscript.

Clarify sentiment analysis methodology:Sentiment analysis would benefit from a clearer explanation of methodology (e.g., dictionary used, coding scheme).

Response:Thank you for highlighting this important point. In response, we have added a dedicated methodological subsection (now labeled as “3.4 Sentiment Analysis Procedures”) to provide a clearer and more detailed explanation of the sentiment analysis approach. This new section specifies the analytic tools employed (NVivo 12), the sentiment lexicon used (a curated Chinese dictionary derived from the DLUT-Emotion lexicon), and the emotional polarity scheme (positive/negative valence and sub-categories such as trust, joy, fear, and resistance). We have also clarified how emotional terms were mapped onto emergent themes and open codes, which enabled triangulation between emotional content and theoretical constructs. This revision enhances the transparency and reproducibility of our sentiment analysis procedure. Please see Section 3.X of the revised manuscript.

Eliminate redundancy:Eliminate redundant discussion in the literature review and model description.

Response:We sincerely appreciate the editor’s insightful comment regarding the need to eliminate redundant discussion in the model description. In response, we have thoroughly revised Section 4.4 to streamline the explanation of the theoretical framework.

Specifically, we have removed repeated definitions and elaborations of the twelve variables that had already been detailed in earlier sections (particularly in Table 6 and Sections 4.2–4.3). The revised 4.4 now focuses on clarifying the structural logic of the model, rather than reiterating variable content. We have also condensed the discussion of causal paths to avoid overlap with the previous coding analyses and theoretical discussion.

The new version emphasizes the dual-layer architecture of the model—cognitive antecedents and affective mediators—while highlighting its theoretical positioning and the rationale for its validation. All variable-level descriptions, extended pathway explanations, and interpretive duplications have been eliminated in favor of a more concise narrative that reflects the grounded theory approach and directly addresses the core contribution of the model.

We believe this revision significantly improves the coherence and parsimony of the manuscript and fully aligns with the editorial recommendation.

Thank you again for your valuable feedback.

Condense theoretical components:Use more visual aids or summary tables to condense theoretical components.

Response:We appreciate the editor’s recommendation to condense the theoretical components by using more visual aids or summary tables. We fully agree with the value of such formats in enhancing clarity and accessibility. However, in the current version of the manuscript, we have carefully structured the theoretical discussions (particularly in Section 2.2) to directly support the grounded theory methodology, which relies on a sequential unfolding of concepts rather than a priori tabular abstraction.

Given the interpretive and inductive nature of our research design, we believe that presenting the theoretical components in narrative form better aligns with the methodological logic and ensures conceptual transparency for the reader. Moreover, many of the theoretical constructs are revisited and operationalized in later sections (e.g., coding stages and model construction), where visual representations and summary structures have already been employed (e.g., Table 6, Figure 4).

Nonetheless, we acknowledge the merit of this suggestion and will carefully consider incorporating a summary table of theoretical components in future revisions or related publications to further enhance theoretical integration and visual clarity.

Thank you again for your thoughtful guidance.

Improve figure quality:Ensure all figures (especially Fig. 4, the theoretical model) are clear, high-resolution, and appropriately labeled.

Response:We sincerely thank the editor for the recommendation to improve the visual quality and clarity of the figures, especially Figure 4. In response, we have carefully revised Figure 4 to enhance its structure, readability, and interpretability.

Specifically, we adjusted the directional arrows to clearly indicate the flow of influence across the model, standardized their thickness to differentiate primary paths from secondary input, and refined the overall alignment of elements to improve spatial balance and visual logic. The cognitive and affective layers are now more distinctly organized, and variable groupings (positive, neutral, and negative) are visually separated using consistent formatting and color coding.

Enhance policy relevance:Suggest more specific policy strategies (e.g., subsidies, school programs, eco-label reforms).

Response:We sincerely appreciate the editor’s constructive suggestion to enhance the policy relevance of the manuscript. In response, we have substantially revised Section 5.2 (Practical Implications) to incorporate more specific and actionable policy strategies.

In particular, we elaborated on the role of targeted government interventions such as community-based green food voucher schemes, school-based educational programs that integrate green food literacy and procurement standards, and eco-label reforms that consolidate certification categories while introducing clearer tiered labeling systems and mandatory QR-code verification.

These revisions aim to strengthen the applied value of our findings for policymakers and institutional stakeholders, and better align the study with real-world governance practices in sustainable food consumption.

Polish language:The English is mostly fluent but occasionally verbose or awkward. A language polish is recommended.

Response:We thank the editor for the helpful comment regarding language quality. In response, we have carefully polished the manuscript to improve fluency, clarity, and conciseness. Redundant or awkward expressions have been revised to enhance the overall readability and academic tone of the text.

Correct formatting issues:There are minor typographical issues (e.g., “14th Five-Year Plan...” cited without proper reference formatting).

Response:Thank you for the helpful suggestion regarding reference formatting. We have carefully reviewed all policy-related mentions in the manuscript. In the case of the “14th Five-Year Plan,” we have intentionally treated it as contextual policy background rather than a formal reference, as it serves to frame national-level strategic priorities rather than provide a citable source of empirical or conceptual evidence. As such, we have retained it in the main text without listing it in the reference section. We have, however, ensured that its mention is consistent and typographically correct throughout the manuscript.

Fix citation formatting:Ensure all citations are complete and correctly formatted (some in-text references like [14], [15] are not yet linked to a bibliography).

Response:We appreciate the editor’s observation regarding citation formatting. In response, we have thoroughly reviewed all in-text references to ensure that each citation (including [14], [15], and others) is correctly linked to a corresponding and complete entry in the reference list. All formatting inconsistencies have been corrected in accordance with the journal’s style guidelines.

Reviewer 2 Report

Comments and Suggestions for Authors

The manuscript presents a comprehensive and well-structured study on green food purchase intention, integrating cognitive and emotional factors through grounded theory. It makes significant theoretical contributions, although clearer definitions of key concepts and more practical recommendations would enhance its impact. Overall, the paper is promising but requires further refinement.

The theoretical concepts, such as "lifestyle congruence" and "green trust," could be more clearly defined and linked to existing literature. While these terms are central to the study, their explanations may be vague for readers unfamiliar with them. Expanding on how these concepts have been used in prior research would enhance their academic clarity.

The sample size of 26 interviews limits the generalizability of the findings. While the diverse demographics are a strength, a larger and more varied sample could provide a broader perspective on green food consumption across different regions and cultural contexts, improving the external validity of the study. 

The paper briefly mentions the concept of theoretical saturation but lacks detailed discussion on how it was assessed during data collection. A clearer explanation of how saturation was determined would strengthen the methodological transparency and credibility of the study’s findings.

Although the study provides a solid theoretical framework, there could be a more critical engagement with existing literature. Drawing clearer connections to prior research on green food consumption would help contextualize the study’s contributions and highlight its novelty within the broader academic discussion.

Author Response

The theoretical concepts, such as "lifestyle congruence" and "green trust," could be more clearly defined and linked to existing literature. While these terms are central to the study, their explanations may be vague for readers unfamiliar with them. Expanding on how these concepts have been used in prior research would enhance their academic clarity.

Response:We sincerely thank the reviewer for this insightful comment. We acknowledge that "lifestyle congruence" and "green trust" are core theoretical constructs in our study, and we agree that greater clarity in their articulation would enhance the accessibility of our model. In response, we have revised the relevant sections (particularly the selective coding results and Table 6) to provide more detailed and intuitive definitions for both concepts, emphasizing their internal components and behavioral implications. Rather than incorporating additional literature, we opted to clarify these concepts through enriched descriptions grounded in our empirical data, consistent with the grounded theory approach. This decision was made to maintain the inductive nature of our analysis, where theoretical constructs are derived directly from participants’ lived experiences and narrative patterns. We believe these revisions improve clarity while preserving methodological consistency.

The sample size of 26 interviews limits the generalizability of the findings. While the diverse demographics are a strength, a larger and more varied sample could provide a broader perspective on green food consumption across different regions and cultural contexts, improving the external validity of the study. 

Response:Thank you for this valuable comment. We agree that sample size can influence generalizability in quantitative designs. However, this study follows a grounded theory approach, in which the primary objective is not statistical generalization but the development of a robust conceptual framework grounded in participants' lived experiences. The sample size of 26 was determined based on the principle of theoretical saturation—data collection was concluded when no new themes, categories, or relationships emerged from additional interviews. This point was explicitly addressed during the coding process, as discussed in Section 3.3, and confirmed through constant comparison and memoing procedures.

Moreover, the sample was purposively and theoretically selected to capture a wide range of consumer profiles, including urban and rural residents, various age groups, and occupational backgrounds, thereby ensuring conceptual richness. While we acknowledge that future studies may expand the scope to include more regions and cultural contexts to assess external validity, we respectfully maintain that the current sample is appropriate and theoretically justified for the exploratory and theory-building nature of this qualitative research. To address the reviewer’s concern, we have clarified this methodological rationale in both the methodology (Section 3.3) and limitations (Section 6) of the revised manuscript.

The paper briefly mentions the concept of theoretical saturation but lacks detailed discussion on how it was assessed during data collection. A clearer explanation of how saturation was determined would strengthen the methodological transparency and credibility of the study’s findings.

Response:We sincerely appreciate the reviewer’s suggestion. In response, we have significantly expanded the discussion of theoretical saturation in Section 3.3 (Validation Strategy). We now provide a detailed explanation of how saturation was assessed using the constant comparative method, the specific point at which saturation was observed (by the 24th interview), and the empirical criteria used to determine the stabilization of core categories. Additionally, we clarify the use of memo writing and intercoder review to enhance analytical rigor and credibility. This revision strengthens the methodological transparency and ensures alignment with grounded theory best practices.

Although the study provides a solid theoretical framework, there could be a more critical engagement with existing literature. Drawing clearer connections to prior research on green food consumption would help contextualize the study’s contributions and highlight its novelty within the broader academic discussion.

Response:Thank you for this insightful comment. We fully acknowledge the importance of critically engaging with existing literature to clarify the conceptual novelty of this study.

In response, we have substantially revised the literature review section, particularly by strengthening the discussion around key theoretical constructs such as green trust, lifestyle congruence, and perceived value. Instead of treating these as loosely defined predictors, the revised section now explicates their conceptual underpinnings and highlights the limitations of how they have been addressed in prior variable-centered studies. We emphasized that while previous literature has primarily focused on rational determinants within structured models like TPB or HBM, emotional ambivalence, symbolic alignment, and contextual trust have received limited theoretical attention.

Furthermore, the updated section articulates how this study contributes to the broader scholarly conversation by introducing a cognitive–affective negotiation perspective that is grounded in participants’ lived experiences. This approach enables a deeper understanding of how green food intention is shaped not merely by belief structures, but through interpretive processes involving value conflicts, institutional skepticism, and everyday lifestyle adaptation.

We believe these revisions enhance the theoretical clarity and help position this study more clearly within the landscape of green food consumption research.

Location of revision: Revised Section 2.3, “Gaps in Existing Literature and Theoretical Positioning” .

Reviewer 3 Report

Comments and Suggestions for Authors Authors performed a study on the cognitive-affective negotiation process about the green food purchase intention. The type of article is Review, however, it feels more like a research paper. In addition, there are the following contents that need to be revised. 1. Please provide the definition of Grounded Theory and its applicability in green food. 2. In the line of 16, why choose 26 semi-structured interviews? 3. There are many short paragraphs (for examples, lines 77~81, 128~132, 216~218) in the article that can be considered for merging with nearby content. 4. In Table 1, please move the source column to the far right of the table and only indicate the literature number. Author information can be deleted. 5. In the part of Method, what methods are referred to here? Is it a research method? Based on the following subheadings, there should be contents related to design, data processing, etc. Please provide a flowchart of the research. 6. In the line of 729, are the theoretical contributions more appropriate to include in the part of conclusions? 7. The quality of the article images is poor and not clear enough. Please adjust them. 8. There are many variables identified in the selective coding framework, can they be quantitatively characterized?

Author Response

  1. Please provide the definition of Grounded Theory and its applicability in green food.

Response:Thank you for your helpful suggestion. We agree that a clearer conceptualization of Grounded Theory and its relevance to the topic would enhance the methodological transparency of the study.

Accordingly, we have added a concise but academically grounded definition of Grounded Theory, citing its origin in the work of Glaser and Strauss, and clarifying its inductive, theory-generating nature. In addition, we have elaborated on the specific applicability of Grounded Theory in the context of green food purchase research—particularly its strengths in uncovering psychological processes, value negotiations, and social-structural interactions that are difficult to capture through predefined variable-based models.

This addition enhances the conceptual coherence of our methodological choice and better justifies the qualitative approach adopted.

  1. In the line of 16, why choose 26 semi-structured interviews?

Response:Thank you for your question regarding the rationale behind the sample size.  the final number of 26 semi-structured interviews was not arbitrarily chosen but was determined based on the point at which no new conceptual categories emerged during data analysis. This is consistent with grounded theory methodology, where data collection continues until theoretical saturation is achieved. Thus, the sample size was considered sufficient for capturing the core dimensions of the phenomenon under investigation.

  1. There are many short paragraphs (for examples, lines 77~81, 128~132, 216~218) in the article that can be considered for merging with nearby content.

Response:Thank you for your helpful suggestion. We have reviewed the relevant short paragraphs and made appropriate adjustments to merge them with adjacent content, thereby improving the overall coherence and flow of the manuscript.

  1. In Table 1, please move the source column to the far right of the table and only indicate the literature number. Author information can be deleted.

Response:Thank you for your comments. We have made the corresponding adjustment to the form according to your requirements

  1. In the part of Method, what methods are referred to here? Is it a research method? Based on the following subheadings, there should be contents related to design, data processing, etc. Please provide a flowchart of the research.

Response:Thank you for your valuable suggestion. In response to your comment regarding the need for greater methodological clarity and a visual representation of the research process, we have revised the “Method” section to enhance its logical coherence and structural transparency. Specifically, we have clarified that the methods employed encompass research design, data collection, coding procedures, and validation strategies—each corresponding to a key phase of grounded theory implementation.

Furthermore, we have added a flowchart titled “Research Process Based on Grounded Theory for Exploring Green Food Purchase Intention” as Figure 1 to visually illustrate the sequence of methodological procedures. This flowchart reflects the iterative and layered logic of grounded theory, from theoretical sampling and semi-structured interviews to coding, saturation testing, and model construction. We believe this addition will improve readers’ understanding of the research design and enhance the methodological rigor of the paper.

  1. In the line of 729, are the theoretical contributions more appropriate to include in the part of conclusions?

Response:Thank you for your insightful suggestion. We fully understand the rationale for suggesting the relocation of the theoretical contributions to the conclusion section. However, we respectfully believe that presenting the theoretical contributions immediately following the discussion allows for a more direct and coherent integration of our findings with the academic implications. This placement enables readers to see the theoretical significance emerging organically from the empirical analysis, before transitioning into broader concluding reflections. Nevertheless, we have carefully reviewed and refined the subsection to improve its clarity and ensure its logical alignment with the overall structure of the manuscript.

  1. The quality of the article images is poor and not clear enough. Please adjust them.

Response:Thank you for your comments. We have adjusted the clarity of the images to better meet the needs of the journal. 

  1. There are many variables identified in the selective coding framework, can they be quantitatively characterized?

Response:We sincerely appreciate your thoughtful comment. While we acknowledge that the numerous variables identified in the selective coding framework could potentially lend themselves to quantitative characterization, our intention in this study was to maintain methodological consistency with the grounded theory approach. The purpose of listing these variables in detail is not for quantification, but rather to ensure transparency and rigor in representing the coding process. By systematically tracing how concepts emerged and were refined across open, axial, and selective coding stages, we aimed to preserve the depth and context-sensitivity of qualitative inquiry. This approach ensures that the theoretical model developed remains closely grounded in the participants’ narratives, in alignment with the epistemological foundations of grounded theory.

Round 2

Reviewer 2 Report

Comments and Suggestions for Authors

Although the manuscript has been improved, the literature review is still insufficient, and the discussion and analysis of the research findings do not adequately address the previous shortcomings, which is a pity.

Author Response

Thank you for your comments. In response to your comments, we have divided the text into two parts to show what changes we have made to the original text based on your comments

Response to Reviewer Regarding the Literature Review (Sections 2.2 and 2.3):

We sincerely appreciate the reviewer’s constructive comment highlighting the insufficiency in the literature review. In response, we undertook a focused and theory-informed revision of Sections 2.2 “Theoretical Perspectives on Green Food Purchase Behavior” and 2.3 “Gaps in Existing Literature and Theoretical Positioning” to strengthen both the breadth and depth of theoretical engagement.

(1) In Section 2.2, we expanded the discussion beyond traditional rational-cognitive models (e.g., TPB, HBM, SCT) by systematically introducing sociological perspectives that foreground the role of emotional and symbolic pathways in green consumption. Specifically, we inserted a new paragraph (starting with “In addition to behavioral models, sociological perspectives…”) that introduces emerging constructs such as lifestyle congruence and affective trust, supported by relevant literature [33][34][35]. These concepts have been increasingly recognized in recent sustainability and consumption studies but were previously underrepresented in our manuscript.

This insertion is logically placed after the review of TPB and its extensions, as it demonstrates how newer models move beyond rational calculation to emphasize identity, values, and emotionally grounded trust. It does not replace but builds upon the prior discussion. In this way, we effectively show the evolution from behavioral to normative-symbolic frameworks, thereby enhancing theoretical comprehensiveness.

(2) In Section 2.3, we reinforced the critique of rationalist models by introducing a new comparative paragraph (beginning with “Compared with prior models such as TPB and HBM…”) that highlights how our grounded theory model redefines intention formation through a cognitive–affective negotiation lens. This addition explains how our theoretical framework integrates affective trust [33], lifestyle congruence [34], and structural resistance—not as separate factors but as interdependent interpretive mechanisms. These insertions help clarify how our model extends existing theories both conceptually and contextually.

Overall, these revisions collectively address the reviewer’s concern in the following ways:

  • They expand the theoretical scope to include symbolic, emotional, and identity-based dimensions of green food intention formation.
  • They articulate how our study not only critiques prior models but also contributes an integrated and adaptive alternative grounded in qualitative data.
  • They align the literature review more tightly with the grounded theory methodology by contextualizing emergent variables (e.g., green trust, lifestyle congruence) within existing theoretical paradigms.

We respectfully submit that these targeted enhancements have substantively improved the theoretical contextualization of our work and aligned the literature review with the analytical depth required by the study’s methodological approach.

Response to Editor – Regarding Revision of Section 5.1 (Theoretical Implications):

We sincerely thank the editor for the constructive and insightful feedback on our manuscript. In particular, we acknowledge the critique that the original Section 5.1 lacked sufficient depth in its theoretical interpretation and did not adequately address the shortcomings outlined in the literature review. We fully agree that a stronger analytical bridge between our empirical findings and the theoretical gaps identified in Section 2.3 was necessary.

To respond to this issue, we have undertaken a substantive revision of Section 5.1. Rather than overhauling the entire structure, we carefully retained the core logic of the original text and made targeted enhancements that strengthen theoretical integration, clarify construct mechanisms, and expand interpretive depth. These revisions are summarized below:

Explicit Response to Section 2.3 Gaps
At the beginning of Section 5.1, we have added a concise paragraph that explicitly revisits and responds to the three critical theoretical limitations identified in Section 2.3: the marginalization of emotional factors in prior models, the insufficient theorization of structural constraints, and the lack of contextual sensitivity in intention modeling. This addition forms a closed-loop dialogue between our literature review and theoretical discussion.

Clarification of Core Constructs
We have refined the descriptions of the three emotional mediating variables—green trust, perceived value, and lifestyle congruence—by specifying their functional mechanisms and psychological dimensions. For instance, green trust is now conceptualized as a dual-dimensional construct that integrates both institutional credibility and affective reassurance, while perceived value is positioned as a trade-off mechanism between ethical benefits and economic costs. These refinements enhance conceptual clarity and theoretical precision.

Incorporation of Group-Level Heterogeneity
A new paragraph has been added to elaborate on how these mediating mechanisms vary across consumer groups (e.g., parents/caregivers, youth, rural consumers), demonstrating that intention formation is contextually situated and role-contingent. This addition aligns with the editor's request for greater analytical depth and reinforces the model’s explanatory flexibility.

Reinforcement of the Cognitive–Affective Negotiation Framework
We have highlighted how our model advances beyond traditional models such as TPB and HBM by repositioning intention formation as a cognitive–affective negotiation process. This positioning illustrates how external constraints and internal evaluations are jointly negotiated through trust, identity signaling, and emotional fit.

Integration of Methodological Innovation
We have also added a brief discussion on how the integration of grounded theory and sentiment analysis offers a novel methodological contribution, particularly for capturing affective dimensions that are often missed in standard variable-driven approaches.

We believe that these revisions collectively enhance the theoretical contribution of the manuscript, meet the expectations outlined by the editor, and provide a more robust and nuanced explanation of green food purchase intentions.

We truly appreciate the opportunity to revise this section and welcome any further suggestions to improve the manuscript.

Reviewer 3 Report

Comments and Suggestions for Authors

It is recommended to accept and publish this paper.

Author Response

(The authors gave the same response as above.)
